# Chronic stress causes striatal disinhibition mediated by SOM-interneurons in male mice

Diana Rodrigues [1,2], Luis Jacinto [1,2], Margarida Falcão[1,2], Ana Carolina Castro[1,2], Alexandra Cruz[1,2], Cátia Santa[3], Bruno Manadas [3], Fernanda Marques[1,2], Nuno Sousa [1,2] & Patricia Monteiro [1,2] ✉

Chronic stress (CS) is associated with a number of neuropsychiatric disorders, and it may also contribute to or exacerbate motor function. However, the mechanisms by which stress triggers motor symptoms are not fully understood. Here, we report that CS functionally alters dorsomedial striatum (DMS) circuits in male mice, by affecting GABAergic interneuron populations and somatostatin positive (SOM) interneurons in particular. Specifically, we show that CS impairs communication between SOM interneurons and medium spiny neurons, promoting striatal overactivation/disinhibition and increased motor output. Using probabilistic machine learning to analyze animal behavior, we demonstrate that in vivo chemogenetic manipulation of SOM interneurons in DMS modulates motor phenotypes in stressed mice. Altogether, we propose a causal link between dysfunction of striatal SOM interneurons and motor symptoms in models of chronic stress.

Stress response is an allostatic mechanism triggered by a perceived stressor. This natural adaptive response helps organisms to cope with and overcome stressful situations and is largely mediated by the HPA (hypothalamic-pituitary-adrenal glands) axis[1–3]. However, repeated exposure to stressful events leads to chronic HPA overactivation with detrimental consequences at several body systems[4–10]. Chronic stress (CS) affects brain structure and function, and several studies have shown that one of the brain regions highly impacted by CS is the striatum, leading to impairments in striatal-mediated behaviors such as motor and action planning[11–18]. Accordingly, CS is a well-recognized risk factor for a number of neuropsychiatric disorders, namely obsessive-compulsive disorder (OCD; characterized by repetitive motor routines)[19,20], posttraumatic stress disorder (PTSD; characterized by hyperarousal and exaggerated startle reactions)[21,22], anxiety[23,24], and depression[25–28], but it also worsens symptom progression in patients with Parkinson's and Huntington's disease[29,30], two striatal motor disorders. Stress also exacerbates motor symptoms in OCD[19,31], a disorder with increased metabolic activity in the human caudate nucleus[32,33], which corresponds to rodents' dorsomedial striatum[34,35]. But despite evidence of striatal dysfunction upon CS exposure, its

neurobiological consequences at cellular, molecular, and synaptic levels, and how these can contribute to neuropsychiatric motor symptoms remain elusive. Uncovering the pathway by which stress triggers motor dysfunction might be relevant to explain why motor symptoms emerge/worsen in multiple brain disorders after chronic stress exposure.

Here, by combining neuroproteomics, electrophysiology, optogenetics, chemogenetics, and probabilistic machine learning for analyzing animal behavior, we demonstrate that CS has a detrimental impact on GABAergic striatal interneurons, altering the functional neuronal dynamics of striatum circuits in male mice. Mechanistically, we found that CS leads to striatal overactivation by reducing the connectivity strength between GABAergic SOM interneurons and medium spiny neurons (MSNs). Unsupervised behavioral phenotyping of chemogenetically manipulated animals further revealed that in vivo chemogenetic modulation of striatal SOM interneurons is sufficient to modulate stress-induced motor behaviors. Our findings suggest that targeting striatal SOM interneurons might provide an alternative therapy for treating stress-related motor symptoms.

[1]Life and Health Sciences Research Institute (ICVS), School of Medicine, University of Minho, Braga, Portugal. [2]ICVS/3B's-PT Government Associate Laboratory, Braga/Guimaraes, Braga, Portugal. [3]CNC-Center for Neuroscience and Cell Biology, University of Coimbra, 3004-504 Coimbra, Portugal. ✉e-mail: pmonteiro@med.up.pt

## Results

### Chronic-stress-induced physiological phenotypes

We started by exposing male mice to different daily stressors, in a variable manner, to mimic the unpredictability and variability of stressors encountered in daily life[12,36,37]. This protocol of chronic unpredictable stress aims to avoid the resilient effect of behavioral control over stressors[12,36,37]. After 21-days of chronic stress exposure, we examined individual body weight, adrenals, and thymus of all mice, to assess the consequences of chronic HPA axis activation. As expected, we found that mice in the stressed group presented reduced body weight gain (Fig. 1a), increased adrenal glands weight (Fig. 1b) and decreased thymus weight (Fig. 1c) (Supplementary Fig.1). Moreover, the stressed mice group exhibited significantly increased locomotion (total distance traveled) when placed in an open field (OF) arena and were more likely to avoid its brightly lit center area when compared to littermate non-stressed control mice (Fig. 1d). These results indicate that animals developed physiological alterations consistent with chronic activation of the HPA axis. However, results also showed great individual heterogeneity. In other words, the simple categorization of tested animals into "stressed" and "non-stressed" groups did not fully account for the physiological and behavioral diversity that emerged after CS exposure, thus requiring a large experimental sample size to reveal the effects of CS *per* group. To better evaluate the effect of CS on individual animals, we took advantage of our large sample size to generate receiver operating characteristic (ROC) curves, an objective method used in clinical epidemiology for binary classifiers[38–40]. This approach has also been recently used to classify animals as resilient or vulnerable to stress based on the outcome of three different behavioral tests[41]. Here, we modified such approach by using individual physiological phenotypes, rather than behavioral testing, as inputs for the classification method. By calculating cutoff values based on Youden J index[41,42], a value of 3.565 gr for body weight gain, $1.225 \times 10^{-4}$ gr for adrenal glands weight and $1.551 \times 10^{-3}$ gr for thymus weight were determined as the cutoff values holding the highest true-positive and lowest false-positive rates for each physiological parameter (Fig. 1e–g). Based on these three values, mice within each group (control and stress) were assigned an individual score between zero and three (D-score). If adrenal values were above the cutoff, body weight gain was below the cutoff and thymus was below the cutoff, a score of 3 was cumulatively assigned to that individual mouse (1 score-value *per* parameter). Each mouse was then classified according to their individual D-score (Fig. 1h, i): non-stressed mice (D-score 0 or 1, if no physiological alterations or just 1 parameter altered) and stressed mice (D-score 2 or 3, if all physiological parameters were altered or just 1 parameter non-altered).

Using our D-scoring system we observed that, similarly to humans[43,44], mice displayed clear individual differences in their physiological response to CS, with 78% of animals in the control group not showing any alteration or just one parameter altered ("true" controls) and 62% of mice in the stressed group showing 2 or 3 parameters altered ("true" stressed mice) (Fig. 1j, k). One approach that has been reported in the literature to solve such biological heterogeneity has been to quantify circulating levels of corticosterone (CORT) to evaluate stress levels in rodents. In this study, however, CORT level was not a good predictor of stress with its ROC curve closely matching the chance level curve (Supplementary Fig. 2). The advantage of the D-scoring system proposed here is that although each of the individual physiological parameters was not correlated with the others, combining them together to attribute a D-score helped improving classification of control vs stress animals (Supplementary Fig. 3). We, therefore, proceeded only with animals from the stress group if they had a D-score of 2 or 3 ("true" stressed mice), and from the control group if they had a D-score of 0 or 1 ("true" controls). No animals have been reassigned/changed group or have been included in any of the subsequent analyzes if not

meeting the above criteria. Accounting for the heterogeneity of CS-induced physiological phenotypes may be crucial to dissect and resolve the neurobiological basis of CS-induced behavioral phenotypes, allowing us to then study its underlying neurobiological mechanisms. Next, we used this approach to deconstruct the cellular, molecular, and synaptic impact of chronic stress on striatal circuits.

### Striatal neuroproteomic changes associated with chronic unpredictable stress implicate GABAA receptors and GABAergic Interneurons

To begin dissecting the molecular impact of CS on striatal circuits, we first performed an unbiased quantitative proteomic analysis of striatal samples from control (D-score 0 and 1) and stressed mice (D-score 2 and 3) (Fig. 2a). Using a SWATH-MS approach (Sequential Window Acquisition of all Theoretical Mass Spectra), we were able to confidently identify and quantify 2328 striatal-expressed proteins. Genome-wide overview and pathway analysis on Reactome database[45] revealed that 160 proteins out of 2328 were over-represented in neuronal systems' pathways (Fig. 2b). When looking at the fold enrichment of these 160 neuronal proteins between control and stress samples, we found that the top ten down-regulated proteins by CS included five proteins typically expressed by GABAergic interneurons (gamma-amino butyric acid (GABA) receptor α3, β1, and γ2, Shank1, DLGP1)[46–51], and four proteins related to inhibitory synapses (GABA receptor α3, β1, and γ2, SLIK5)[52] (Fig. 2c). These results suggest a strong convergence on striatal inhibitory pathways and GABAergic interneurons. Because GABAergic interneurons are major gatekeepers of striatal output and are known to exert powerful control over striatal principal neurons[53], we decided to investigate the consequences of CS over striatal circuits at the functional level.

### Chronic stress impairs striatal interneurons and causes persistent abnormal overactivation of striatum

Over 90% of striatal neurons are MSNs, which receive excitatory inputs mainly from the cortex and thalamus, and inhibitory inputs mainly from local GABAergic interneurons[34,54,55]. Several studies have demonstrated that GABAergic interneurons tightly regulate MSNs, and interfering with interneurons' activity can significantly impact the function of entire striatal circuits[56–59]. Given our proteomics data suggesting that CS may be affecting GABAergic interneurons, we decided to perform electrophysiological recordings in anesthetized mice to evaluate the functional impact of CS on striatal interneurons in vivo. However, because striatum encompasses a large brain area with distinct dorsal, ventral, lateral, and medial striatum territories, we wondered whether CS impacts the entire striatum or any striatal territory in particular. To answer this question, we performed brain immunohistochemical quantification of FosB/ΔFosB after CS. Contrary to its close family member c-Fos that accumulates inside neurons that have been acutely activated ~1.5 h ago, FosB/ΔFosB is a very stable member of the Fos-transcription factors that accumulates inside neurons that have been repeatedly activated over several days[60,61]. Upon confocal microscopy examination, we observed that the dorsomedial striatum (DMS) region (homologous to the human caudate nucleus) presented a strong FosB/ΔFosB fluorescent signal. Compared to control littermates, mice repeatedly exposed to stress showed a marked increase in the number of FosB/ΔFosB positive cells (Fig. 3a, b) and displayed higher FosB/ΔFosB fluorescent intensity *per* cell (Fig. 3a, c, d) in the DMS region. Furthermore, approximately 90% of the FosB/ΔFosB positive cells also colocalized with NeuN, a neuronal nuclear antigen commonly used as a biomarker for neurons versus glial cells (Supplementary Fig. 4). Next, we performed in vivo electrophysiological recordings in DMS (Fig. 3e) to check if CS

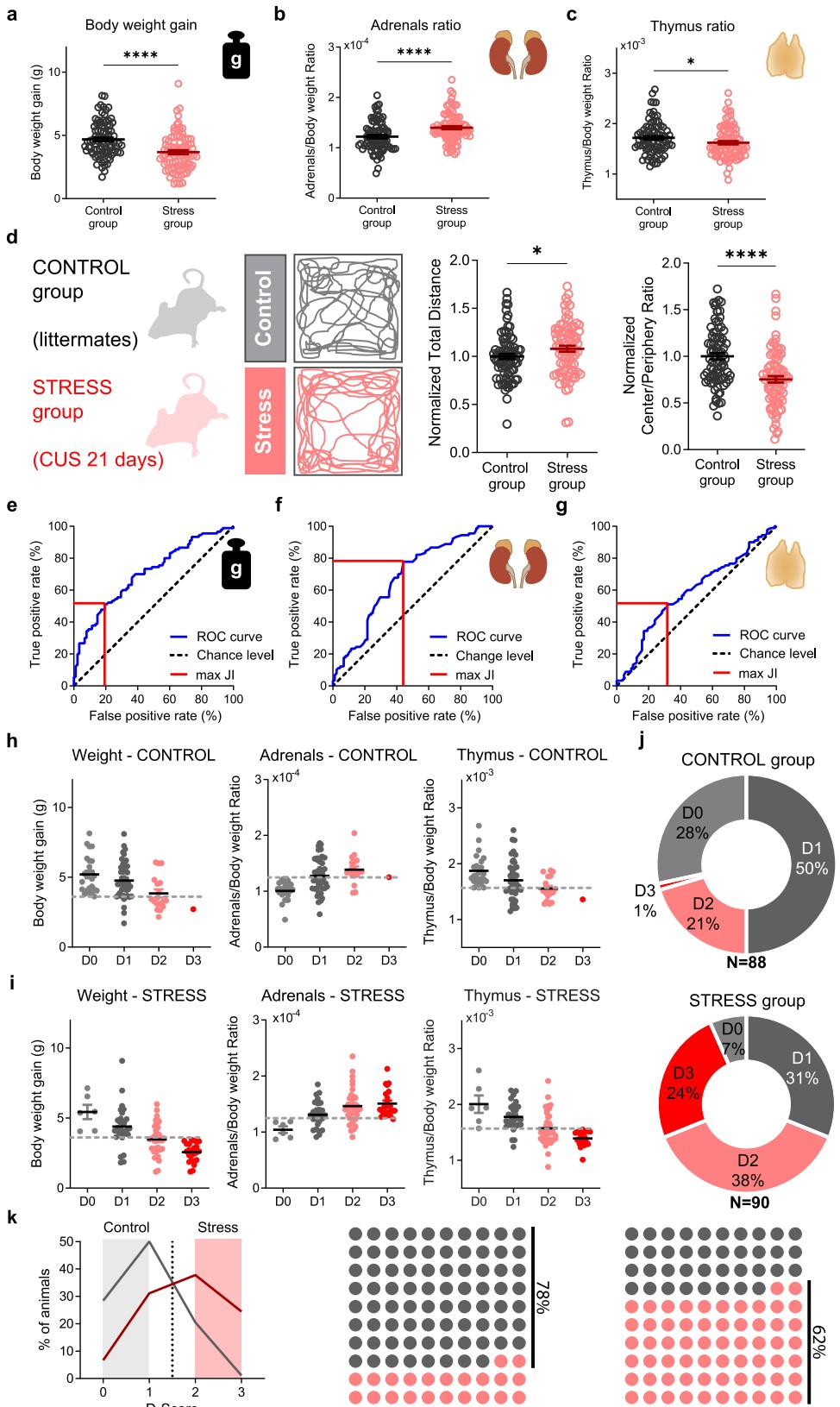

would be affecting striatal circuits at a functional level and impacting GABAergic interneurons in particular, as suggested by our neuroproteomics results. Based on features of spike waveforms (spike half-width and trough-to-peak time; Fig. 3f), 95% of all recorded single units both in control and stressed mice were classified as MSNs (Fig. 3f, g), presenting typical long-duration spike

waveforms. The remaining single units represented ~5% of total recorded units being consistent with striatal interneurons (Fig. 3f, g). After sorting all single units according to both cell types, our data revealed that interneurons displayed reduced average firing rates in stressed mice (Fig. 3h), suggesting that CS is impairing striatal interneurons function. Inversely, the same

**Fig. 1 | Classification of chronic-stress-induced physiological phenotypes.**
**a** Body weight gain for control (black) and stress (red) mice. Data are mean ± SEM; $n = 88$ control and $n = 90$ stress mice; two-sided Welch's unpaired t-test ****$p < 0.0001$. **b** Adrenal glands weight normalized for body weight of each control (black) and stress (red) mouse. Data are mean ± SEM; $n = 88$ control and $n = 90$ stress mice; two-sided Welch's unpaired t-test ****$p < 0.0001$. **c** Thymus weight normalized for body weight of each control (black) and stress (red) mouse. Data are mean ± SEM; $n = 88$ control and $n = 90$ stress mice; two-sided Welch's unpaired t-test *$p = 0.0451$. **d** Left: illustration of control and stress mice trajectories in the open field arena (OF); Middle: Normalized total distance traveled in the OF over 10 min; Right: Normalized ratio of total distance traveled in central over peripheral part of the OF arena for 10 min. Data are mean ± SEM; $n = 82$ control and $n = 83$ chronic stress mice; two-sided Welch's unpaired t-test *$p = 0.049$, ****$p < 0.0001$. Credit: image was adapted from https://scidraw.io. Receiver operating characteristic (ROC) curves (blue line) for body weight gain (**e**), adrenal

glands (**f**), and thymus (**g**), maximum Youden J index (JI; red line). Dashed line represents chance level. Body weight gain, adrenal glands, and thymus of control (**h**) and stressed mice (**i**) with zero (gray), one (dark gray), two (pink) or three (red) altered physiological parameters. Dashed line corresponds to the cutoff value (data represent means ± SEM; $n = 88$ control and $n = 90$ stress mice). **j** Percentage of the total population of control (top) and stress (bottom) mice with zero (gray), one (dark gray), two (pink) or three (red) altered physiological parameters. **k** Left: D-score histograms of control (gray) and stress (red) mice. Dashed line represents the score from where the percentage of stress mice starts to dominate in the total population (D-score > 1); Right: 10 × 10 Dot plots of control and stress mice: 78% of animals in the control group showed only 1 parameter altered or none ("true" controls, D0 + D1 mice, gray) and 62% of mice in the stressed group show 2 or 3 parameters altered ("true" stressed mice, D2 + D3 mice, red). Source data are provided as a Source Data file.

electrophysiological recordings showed increased MSNs' firing rate in stressed mice, revealing a scenario of reduced firing from striatal interneurons with concomitant disinhibition/overactivation of MSNs (Fig. 3i). Together these findings suggest loss of interneurons inhibitory control over MSNs, leading to abnormal sustained activation of striatal circuits after chronic stress.

## Chronic stress impairs synaptic communication between SOM interneurons and medium spiny neurons

Our combined findings from proteomic analysis and in vivo electrophysiology recordings show that CS is affecting inhibitory transmission and GABAergic interneurons. The striatum contains two major populations of inhibitory GABAergic interneurons: the parvalbumin- (PV) and somatostatin- (SOM) positive interneurons. To dissect which population is being affected by CS, we performed targeted ex vivo patch-clamp electrophysiology recordings from both PV (Fig. 4a) and SOM interneurons (Fig. 4f). In order to correctly identify both cellular populations, PV-tdTomato mice were used as well as SOM-Cre mice crossed with Rosa-tdTomato conditional reporter mice (SOM-cre:tdTomato mice). This strategy allowed cell-type specific recordings from tdTomato fluorescent cells located in the DMS region. Recordings of miniature inhibitory postsynaptic currents (mIPSC) revealed that stressed mice exhibit higher frequency in PV interneurons (Fig. 4b) and lower mIPSC frequency in SOM interneurons (Fig. 4g), without major differences in amplitude (Fig. 4c, h). This indicates that the effects of CS on striatum are cell-type specific, likely causing an upregulation in the total number of inhibitory synapses in PV interneurons and an opposite decrease in SOM interneurons. We also observed an increase in mIPSC decay in PV interneurons (Fig. 4d) but not in rise-time (Fig. 4e) nor in SOM interneurons mIPSC kinetics (Fig. 4i, j). Such differences in mIPSC decay without affecting rise time can be due to different synaptic composition of GABA-A receptor subtypes, as already suggested by our proteomics results. However, despite these apparent synaptic defects after exposure to CS, it is possible that these synaptic changes do not affect PV and SOM interneurons' final output and inhibitory control over MSNs. To examine this possibility, we performed an additional series of acute brain slice electrophysiological experiments where channelrhodopsin-2 (ChR2) was expressed either in PV or SOM interneurons in order to directly evoke and evaluate their output over MSNs. To avoid issues related to different levels of ChR2 expression, we used a genetic targeting strategy rather than viral-mediated ChR2 expression. PV-Cre and SOM-Cre mice were crossed with Ai27D mice, which express ChR2/tdTomato following exposure to Cre recombinase. This strategy guarantees both cell-type specific expression of ChR2 and reproducible stable levels across different experimental mice while simultaneously providing fluorescent labeling of cells for targeted electrophysiology recordings in DMS. By optogenetically activating either PV or SOM interneurons and simultaneously recording from individual neighboring MSNs, we were able to examine PV-MSN

and SOM-MSN synaptic connectivity strength through optogenetically-evoked postsynaptic currents (oPSC; Fig. 4k, m). Surprisingly, light stimulation of ChR2-expressing PV interneurons in stressed mice did not show differences in the amount of evoked inward current over MSNs when compared to controls (Fig. 4l). In contrast, when stimulating ChR2-expressing SOM interneurons in stressed mice, the peak amplitude and area of striatal oPSC currents was significantly reduced (Fig. 4n), revealing a very prominent alteration in SOM-MSN synaptic connectivity in response to chronic stress.

## In vivo chemogenetic manipulation of striatal SOM interneurons modulates stress-induced motor behavioral phenotypes

To test whether the loss of SOM interneurons control over MSNs could be mediating stress-induced striatal behavioral phenotypes, we performed in vivo chemogenetic viral manipulations of SOM interneurons in control and stress mice (Fig. 5a). Adenovirus expressing DIO-hM3D(Gq)-mCherry was injected into the DMS region of male SOM-Cre mice and allowed to express before the start of the stress protocol. After 21 days of stress exposure, mice were placed in an open arena to examine the impact of chemogenetic activation of hM3D(Gq)-expressing SOM interneurons by intraperitoneal injection of CNO. We found that stressed mice have an hyperlocomotion phenotype when compared with controls (increased total distance traveled) and that activation of SOM interneurons in stressed mice is sufficient to reduce locomotion to control levels (Fig. 5b). However, because naturalistic behavior is more complex than what can be readily detected by human observers, we also performed unsupervised behavioral phenotyping to analyze the latent structure of spontaneous behavior. Using DeepLabCut[62] markerless pose estimates from arena exploration as input for probabilistic machine learning (VAME toolbox - Variational Animal Motion Embedding[63]), we clustered the multivariate signal into 38 behavioral motifs. These motifs were then grouped into communities of motifs with post-hoc coarse behavioral labels such as walk, turn, rear, groom (Fig. 5c). Inspecting the difference in motif usage between SOM-activated and non-activated control and stress mice, revealed several motifs that differed between the groups (Fig. 5d). Of those, eight motifs were grouped into the Walk community (motifs 0, 5, 10, 14, 17, 24, 29, and 34), two in the Turn community (motifs 33 and 37), six in the Rear community (motifs 3, 4, 11, 18, 23, and 30), three in the Stationary rotation community (motifs 16, 20, and 32), eleven in the Stationary exploration/Pause/Sniff community (motif 1, 6, 9, 12, 15, 19, 21, 26, 27, 28, and 36), four in the Groom community (motifs 2, 7, 13, and 31) and one in the Dig community (motif 35) (Fig. 5d). Further comparison of behavioral communities revealed that stressed mice have increased walking that can be restored to control levels upon striatal activation of SOM-interneurons (Fig. 5e). Moreover, we found that self-grooming—a natural behavior known to be suppressed upon exposure to stress[64]—is reduced in CS mice but can be recovered by activating striatal SOM-interneurons (Fig. 5e). To further confirm the

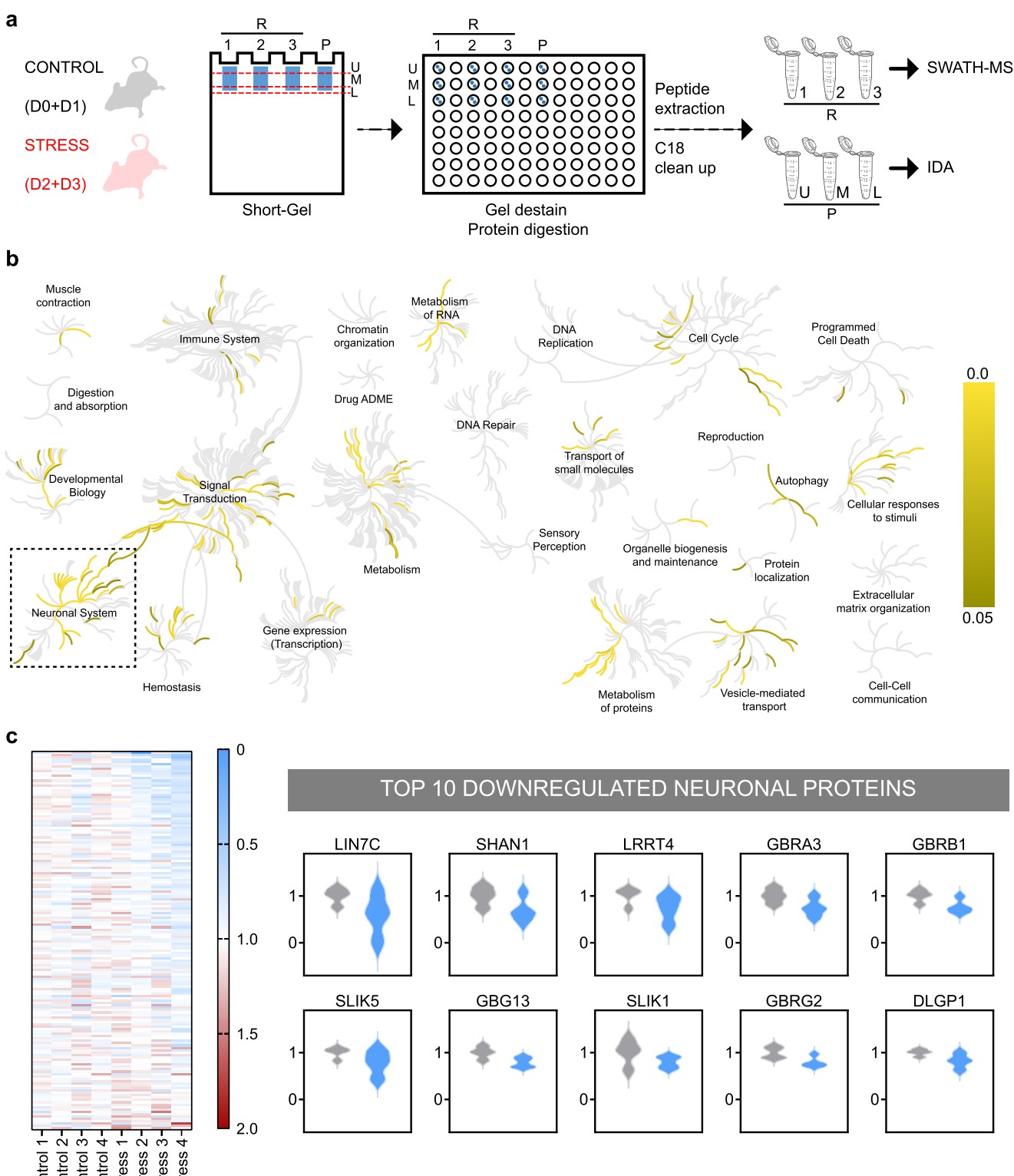

changes in self-grooming detected with clustering analysis of behavioral motifs, we performed the splash test in a cohort of male SOM-hM3Dq control and stress mice, injected with saline or CNO. Once again, we found reduced total grooming time in stress mice compared to controls (Supplementary Fig. 5). A non-significant trend for increased grooming was also observed in the stress group after CNO treatment, but not in the control group (Supplementary Fig. 5). Lastly, to investigate whether locomotion of stressed animals was also affected in a non-anxious environment,

we measured the total distance traveled in the homecage by a group of male SOM-hM3Dq control and stress mice, injected with saline or CNO. In the homecage environment, stressed animals did not seem to display hyperlocomotion phenotype (Supplementary Fig. 6). However, activation of SOM interneurons in the homecage environment suggests a trend to decrease locomotion in stressed mice only, but not reaching statistical significance. Taken together, these results reveal that striatal SOM interneurons are particularly vulnerable to chronic stress, and manipulating their

**Fig. 2 | Liquid chromatography-mass spectrometry (LC-MS) analysis reveals proteomic changes in striatum after chronic stress. a** Experimental design schematic of LC-MS: striatal synaptosomal samples were prepared from control (D0 + D1 mice from the control group) and stressed (D2 + D3 mice from the stress group) animals. All samples were subjected to in-gel digestion after a partial SDS-PAGE run. LC-MS information was acquired in two different acquisition modes: IDA of the pooled samples (P), and SWATH of individual sample replicates (R). U/M/L – upper/middle/lower part of the lane. Credit: image was adapted from https://scidraw.io. **b** Genome-wide overview of hierarchically arranged reactome pathways (input dataset: 2328 proteins identified by LC-MS). Center of each circular "family" is the root of one top-level pathway, for example, "Neuronal Systems". Each branch away from the center represents the next lower level in the pathway hierarchy.

Branch color denotes over-representation of that pathway in the dataset. Light gray branches represent pathways not significantly over-represented. **c** Left: Heatmap view of 160 proteins (y-axis) from the "neuronal system" family of the Reactome database. Protein fold-enrichment is color coded relative to control average (blue: decreased expression; red: increased expression). X-axis represents biological replicates (controls: 1–4; stress: 1–4). Each biological replicate is a combination of 3 brains pooled together: total of 4 control samples obtained from 12 control mice (D0 + D1) and 4 stress samples obtained from 12 stress mice (D2 + D3). Right: Violin plots of top 10 downregulated neuronal proteins in stress mice reveal GABAA receptors and striatal GABAergic interneurons as major targets of chronic stress. Source data is deposited to the ProteomeXchange Consortium via the PRIDE[101] partner repository with the dataset identifier PXD031193.

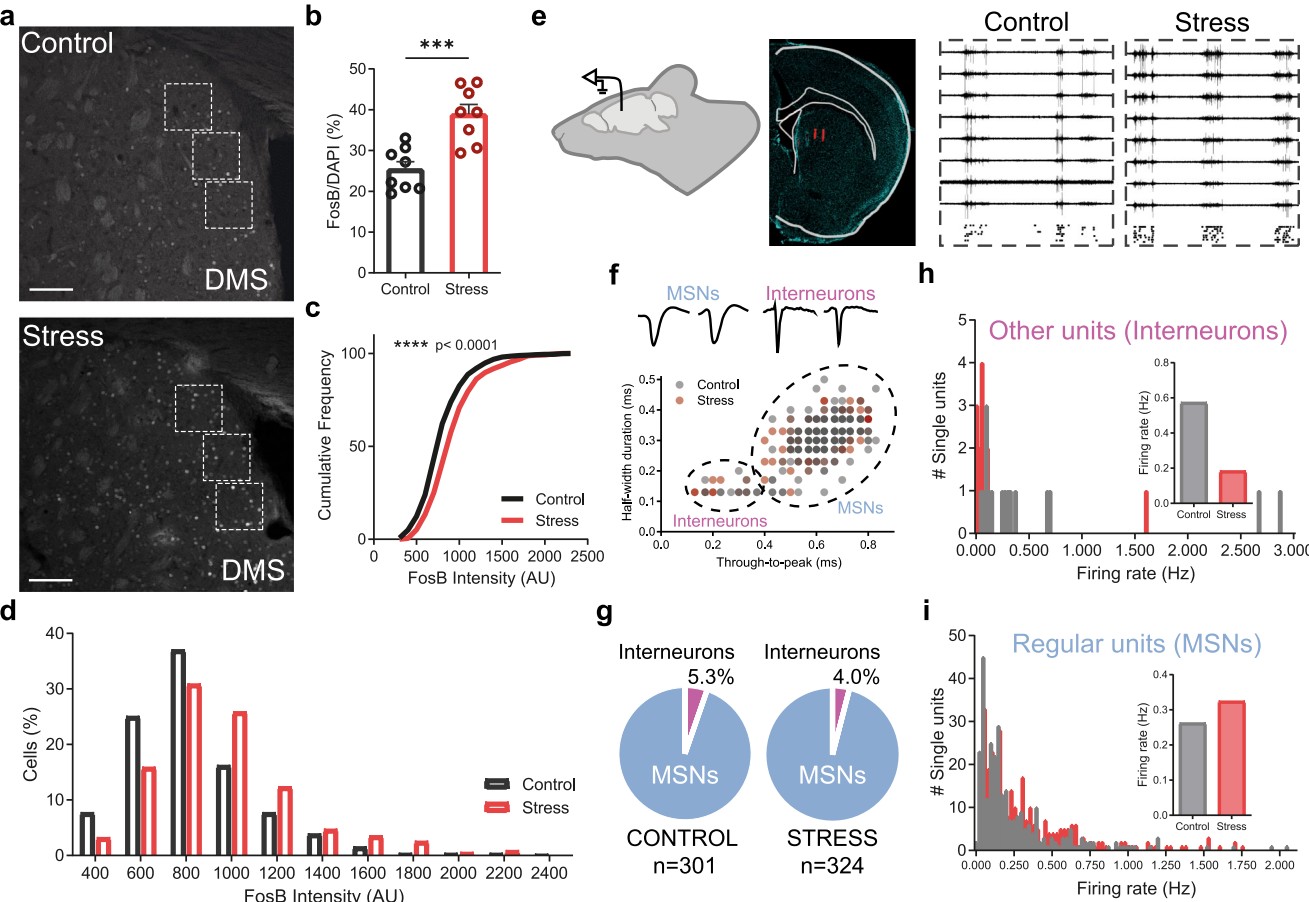

**Fig. 3 | Chronic stress impairs striatal interneurons and causes striatal disin-hibition in vivo. a** Representative immunohistochemistry images of FosB/ΔFosB in dorsomedial striatum (DMS) of control and stressed mice. ROIs for FosB/ΔFosB quantification are represented by dashed white squares. Scale bar = 100 μm. **b** Percentage of FosB/ΔFosB positive cells *per* DAPI nuclei in control and stressed mice (each dot represents one mouse; both hemispheres were quantified in 4 brain slices *per* mouse); Data are mean ± SEM; *n* = 8 control and *n* = 8 chronic stress mice; two-sided Welch's unpaired t-test ***$p$ = 0.0005. **c, d** Cumulative frequency distribution and histogram of FosB/ΔFosB intensity *per* cell in control and stressed mice. Data are mean ± SEM; *n* = 75 cells *per* mice; 8 mice *per* group; Kolmogorov–Smirnov test ****$p$ < 0.0001. **e** In vivo electrophysiology recordings in dorsomedial striatum (DMS) of control and stressed mice. Probe placement in DMS was confirmed by DiI staining (red signal). Representative signal traces from 8 electrodes are shown for control and stressed mice (10 s; signals bandpass filtered

between 0.6 and 6 kHz) with respective raster plots of detected spikes on the bottom. Credit: image was adapted from https://scidraw.io. **f** Scatterplot of single unit's (*n* = 625) spike half-width and trough-to-peak duration with fitted Gaussian Mixture Model clusters classifying MSNs and interneurons from control (gray) and stressed (red) mice; Example spike waveforms of single units classified as medium spiny neurons (MSNs) and interneurons are shown on top. **g** Total percentage of single units classified as interneurons and MSNs in control and stressed mice. *n* = 301 single units recorded in control mice (*n* = 16 interneurons); *n* = 324 single units recorded in stressed mice (*n* = 13 interneurons). **h** Firing frequency histogram and average firing rate of interneurons from control and stressed mice show increased firing in stressed mice during 30 min of spontaneous activity recordings. **i** Firing frequency histogram and average firing rate of MSNs from control and stressed mice show increased firing in stressed mice during 30 min of spontaneous activity recordings. Source data are provided as a Source Data file.

activity is sufficient to modulate motor phenotypes in stressed mice. Future therapeutic interventions that focus on modulating striatal SOM interneurons might provide an alternative therapeutic route for treating stress-related neuropsychiatric disorders affecting striatal circuits.

## Discussion
Our findings demonstrate that chronic exposure to stress leads to overactivation of striatal circuits by reducing the connectivity between GABAergic somatostatin (SOM)-positive interneurons and medium spiny neurons (MSN). One of the major challenges

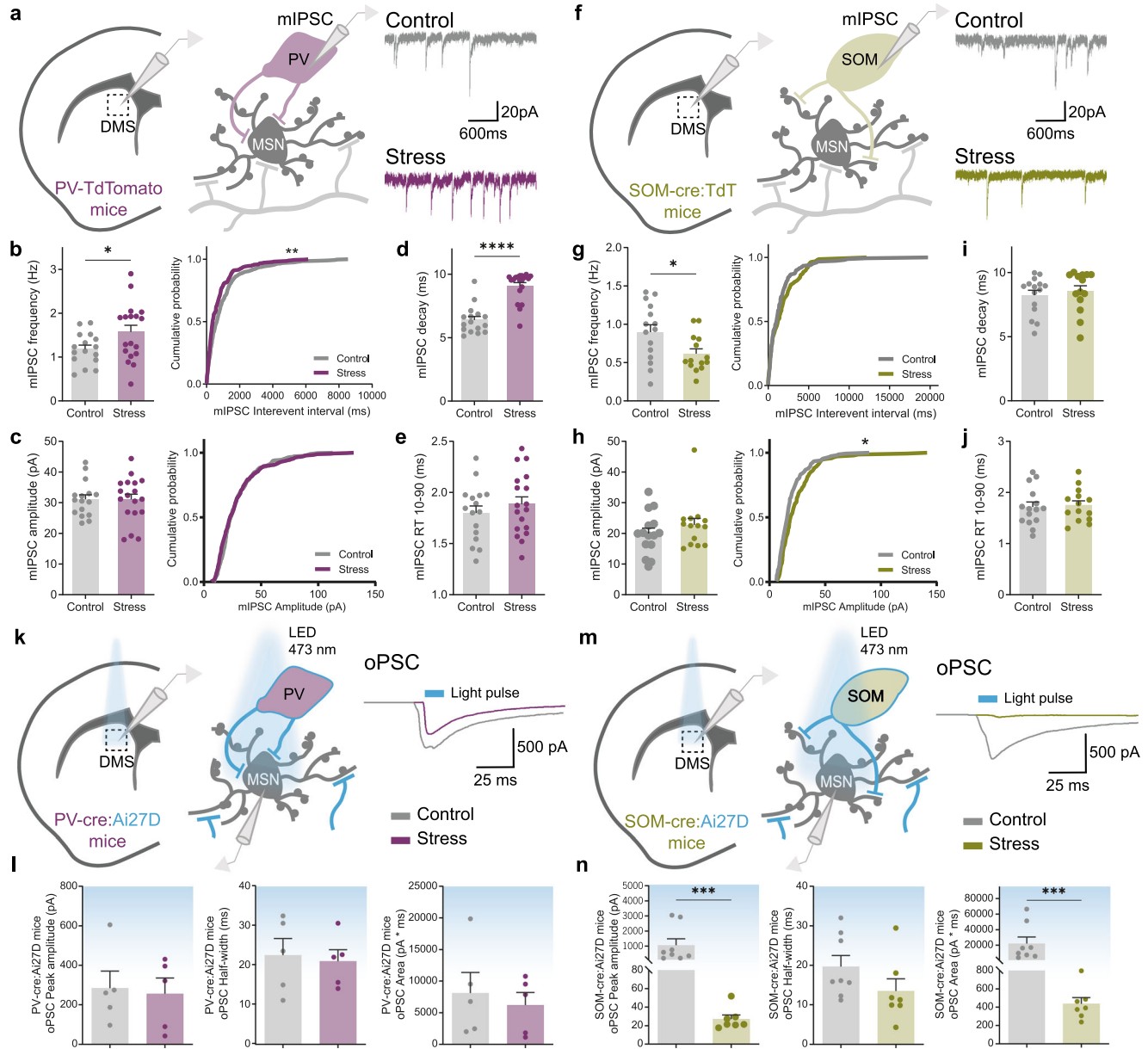

**Fig. 4 | Striatal SOM interneurons are major targets of chronic stress. a** Example traces of miniature inhibitory postsynaptic currents (mIPSCs) recorded from PV interneurons in dorsomedial striatum region (DMS) of control (gray) and stressed (purple) mice. **b** Summary bar graphs (control $n = 16$ and stress $n = 18$ cells; *$p = 0.0387$) and cumulative probability curves (20 events *per* cell; **$p = 0.0019$) show increased mIPSC frequency in PV interneurons of stress mice. **c** Summary bar graphs (control $n = 16$ and stress $n = 18$ cells) and cumulative probability curves (20 events *per* cell) show similar mIPSC amplitude in PV interneurons of stress mice; **d, e** Summary bar graphs (control $n = 16$ and stress $n = 18$ cells) show significantly slower mIPSCs decay kinetics in PV interneurons of stress mice (****$p < 0.0001$), with similar rise times (RT). **f** Example traces of mIPSCs recorded from SOM interneurons in DMS region of control (gray) and stressed (green) mice. **g** Summary bar graphs (control $n = 15$ and stress $n = 14$ cells; *$p = 0.0228$) and cumulative probability curves (10 events *per* cell) show decreased mIPSC frequency in SOM interneurons of stress mice. **h** Summary bar graphs (control $n = 15$ and stress $n = 14$ cells) and cumulative probability curves (10 events *per* cell) show trend for increased mIPSC amplitude in SOM interneurons of stress mice. **i, j** Summary bar

graphs (control $n = 15$ and stress $n = 14$ cells) reveal no difference in kinetics of mIPSC recorded from SOM interneurons of stress mice. **k** Example traces of optogenetically evoked postsynaptic currents (oPSCs) recorded from medium spiny neurons (MSN) upon photostimulation of PV interneurons expressing channelrhodopsin-2 (ChR2) in DMS region of control (gray) and stressed (purple) mice. **l** Summary bar graphs of peak amplitude, half-width, and area, reveal no differences in oPSCs recorded from MSNs upon photostimulation of PV interneurons in stressed mice (control $n = 5$ and stress $n = 5$ cells). **m** Example traces of oPSCs recorded from MSNs upon photostimulation of SOM interneurons expressing ChR2 in DMS region of control (gray) and stressed (green) mice. **n** Summary bar graphs (control $n = 8$ and stress $n = 7$ cells) reveal significantly reduced peak amplitude (***$p = 0.0003$) and area (*$p = 0.0003$) of oPSCs recorded from MSNs upon photostimulation of SOM interneurons in stressed mice. All bar graphs are mean ± SEM. Two-sided Welch's unpaired t-test (**b–e**, **g–j**), Two-sided Mann–Whitney (**l, n**), and Kolmogorov–Smirnov test (**b, c, g, h**). Source data are provided as a Source Data file.

hindering the study of stress is the remarkable variability in individual responses and vulnerability to chronic stress. To address this issue, we developed a physiological-based ROC classification approach based on the work from *Cerniauskas* et al. 2019[41], to

more confidently identify which animals were indeed stressed. Because behavioral assays (previously used by *Cerniauskas* et al. to construct ROC curves) can be highly variable across laboratories (depending on experimental behavioral settings, housing

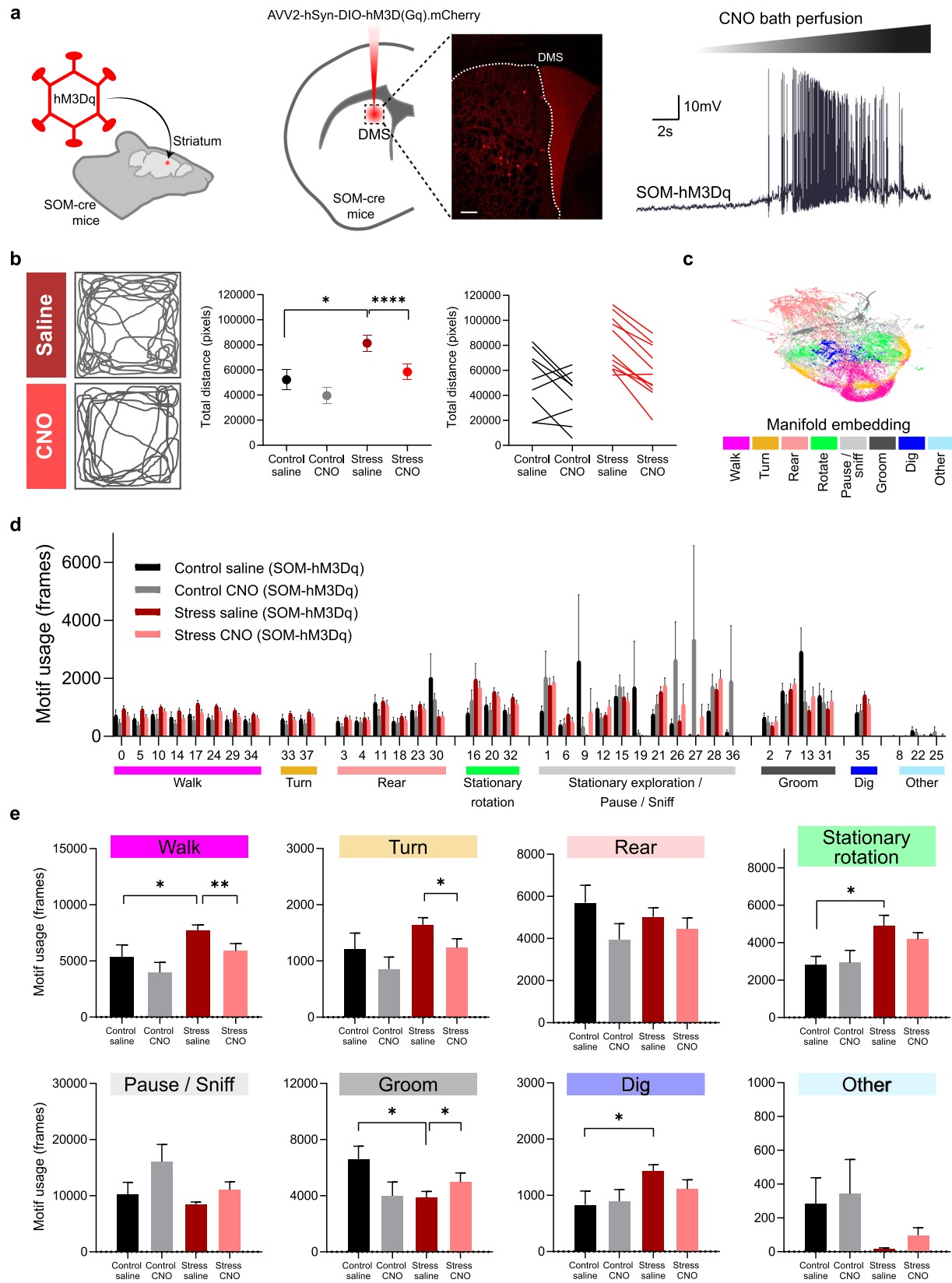

conditions, etc), using physiological parameters to construct ROC curves, as we have done here, may be a more suitable approach to reduce variability, helping to improve the classification method and reproducibility. Leveraging on this classification approach, we then studied the effect of chronic stress (CS) on striatal function and showed that DMS (dorsomedial striatum) is particularly vulnerable to CS, corroborating previous findings from imaging studies performed in stressed rats and humans[18,65].

**Fig. 5 | In vivo manipulation of striatal SOM interneurons in stressed mice reverses stress-induced motor behavioral phenotypes. a** Left: Adenovirus expressing DIO-hM3D(Gq)-mCherry was bilaterally injected in dorsomedial striatum region (DMS) of SOM-Cre mice that were later exposed to chronic stress. Credit: image was adapted from https://scidraw.io. Middle: representative confocal image of *post mortem* brain section of DMS-injected SOM-cre mice showing mCherry (red) viral expression. Scale bar = 200 μm. Right: example trace of whole-cell current clamp recording obtained from mCherry-positive SOM interneuron in DMS slices. Bath perfusion of Clozapine N-oxide (CNO) leads to chemogenetically-induced depolarization and firing in SOM interneurons. **b** Left: illustration of mice trajectories over 10 min in the arena, 1H after administration of saline or Clozapine N-oxide (CNO). Middle: Average total distance traveled in the arena over 10 min for control and stressed mice after saline or CNO injections; Right: Comparison of individual total distance traveled for each mouse with saline and CNO injections. Data are mean ± SEM; *n* = 20 mice; two-sided unpaired t-test *$p$ = 0.0105 and two-

sided paired t-test ****$p$ < 0.0001. **c** Uniform Manifold Approximation and Projection (UMAP) embedding of the behavioral representation encoded by the VAME neural network. Communities of motifs are color-coded. Example UMAP representation from one mouse. **d** Percentage of behavioral motif usage for the control saline (black) and control CNO (gray), and for the stress saline (red) and stress CNO (light red) mice. Data are mean ± SEM; *n* = 20 mice. **e** Percentage of behavioral communities usage for the control saline (black) and control CNO (gray), and for the stress saline (red) and stress CNO (light red) mice. Stressed mice display reduced grooming (*$p$ = 0.0145) and increased walking (*$p$ = 0.0463), digging (*$p$ = 0.0288) and stationary rotations (*$p$ = 0.0157) compared to controls. Chemogenetic activation of SOM-interneurons in stressed mice significantly reduces walking (**$p$ = 0.0093) and turning behavior (*$p$ = 0.0128), while increasing grooming behaviors (*$p$ = 0.0248). Data are mean ± SEM; *n* = 20 mice; saline vs CNO two-sided paired t-test, and control vs stress two-sided Welch's unpaired t-test *$p$ < 0.05. Source data are provided as a Source Data file.

## Vulnerability of striatal GABAergic interneurons to chronic stress

Our proteomic results raise the possibility of GABAergic interneurons being major targets of chronic stress in the striatum. The striatum contains two major interrelated classes of GABAergic interneurons: the parvalbumin (PV) and somatostatin (SOM) positive interneurons. Together these two cell types provide strong inhibitory control of striatal output[56,57] and a tight orchestration between them gives rise to the spatial and temporal properties of local activity[66]. The importance of such cooperation is well illustrated in a recent study whereby using a single interneuron type (PV) in a spiking network model does not allow modulation of excitatory and inhibitory firing rates independently, as it occurs in vivo[66]. The authors demonstrate that strong feedback from SOM interneurons is required for this independent modulation to occur. Thus, disrupting the activity of either SOM or PV has profound consequences on global network firing activity.

In the striatum, SOM and PV interneurons are part of the cortical-striatal circuit: cortex excites both PV and SOM GABAergic interneurons, causing inhibition of local MSNs. Recent work from *Friedman* et al. showed that the cortical-PV excitatory connection is weakened in chronically stressed animals. Furthermore, the authors hypothesize that the PV-MSN inhibitory connection is likely spared[14]. Our results corroborate this hypothesis: by optogenetically activating PV-interneurons and directly recording from connected MSNs, no changes are detected in optogenetically-evoked postsynaptic currents (oPSC) between stress and control animals, suggesting that the PV-MSN connection is indeed intact after chronic stress. By contrast, we show that SOM-MSN connection is severely affected in stressed animals with significantly reduced oPSCs. This suggests that stress differently impacts both interneuron types in the striatum through different mechanisms (cortex-PV *versus* SOM-MSN) but with the same outcome: striatum disinhibition. In other words, stress leads to cortical "disconnection" from PVs (indirectly weakening their inhibitory control over MSNs) and also "disconnects" SOM from MSNs, directly weakening their inhibitory control over MSNs. The two mechanisms potentiate striatum disinhibition under stress.

A causal link between dysfunction of SOM cells and stress-related neuropsychiatric disorders has been previously hypothesized based on rodent studies showing that: (1) chronic stress induces low RNA levels of somatostatin in corticolimbic regions[67]; (2) brain-wide silencing of SOM cells elevates anxiety-like behaviors[68]; and (3) brain wide disinhibition of SOM cells through genetic inactivation of GABAA receptors has antidepressant-like effects[68]. What makes striatal SOM (but not PV) interneurons direct targets of chronic stress is still an open question. In the hippocampus, for example, it has been shown that SOM interneurons express high levels of glucocorticoid receptors (GRs), being well positioned to respond directly to circulating glucocorticoids, while only distinct populations of PV interneurons are direct targets[69]. However, careful consideration should be taken when

extrapolating findings from one brain region to another. Although GR immunoreactivity is present in nearly 90% of striatal cells, SOM interneurons in the striatum do not seem to contain glucocorticoid receptor immunoreactivity[70]. This suggests that stress likely exerts its effects over SOM interneurons through different mechanisms in hippocampus *versus* striatum regions.

Considering such lessons from different brain regions, it is perhaps important to keep in mind that the dorsal striatum itself is composed by two similar yet functionally distinct regions: the dorsolateral "sensorimotor" striatum (DLS) and the dorsomedial "associative" striatum (DMS). Chronic stress causes opposing structural changes in these two regions and promotes a shift from flexible "cognitive" states (DMS encoded) to more rigid "habit" states (DLS encoded)[12,71], rapidly switching the brain from reflective to reflexive control of behavior[72]. Interestingly, a region-specific bias exists in the action of striatal GABAergic interneurons: while PV interneurons control MSN activity more efficiently in the DLS, SOM interneurons more efficiently control MSNs activity in DMS[73]. Furthermore, the relative density of PV expression is known to decrease from DLS to DMS[74,75], which may explain why SOM-MSN connectivity, but not PV-MSN connectivity, is primarily affect in DMS after chronic stress.

## The effect of chronic stress on striatal function and its relevance to disease

Despite scientific evidence linking stress to striatal dysfunction and to motor symptoms worsening/triggering in neuropsychiatric disorders[22,27,28,76], the effect of chronic stress on striatal function remains elusive. Deficits in interneurons' GABAergic transmission, with consequent overactivation/disinhibition of striatum circuits has been proposed as one key mechanism for motor symptoms in OCD and Tourette syndrome[77–82], two disorders known to be affected by chronic stress[19,76,83–85].

Our in vivo electrophysiology recordings reveal reduced activity of local interneurons and abnormally increased MSN activity in DMS (homologous to the human caudate) after chronic stress, mimicking findings of increased caudate activity in OCD patients[32,33] but also corroborating findings from other rodent models of CS[14,86,87].

One of the most replicated findings in human OCD studies is increased activity in the orbitofrontal and anterior cingulate/caudal medial prefrontal cortex, which are also the main sources of input to the striosomal system in the caudate nucleus. Notably, MSNs located at the striosome compartment seem to be particularly vulnerable to chronic stress[14], further suggesting a link between stress, dorsomedial striatum (caudate), and neuropsychiatric disorders[11–18]. A common mechanistic explanation would be that chronic stress releases striatum from the inhibitory influence of local interneurons, promoting unusual motor behavioral output. In our current work, using chemogenetics and probabilistic machine learning to analyze the latent structure of spontaneous animal behavior, the results demonstrate that in vivo

modulation of SOM interneurons in DMS region selectively alters motor activity of chronically stressed male mice.

## Experimental limitations and future perspectives

Despite serving as valuable tools for studying stress physiology, animal models of stress can never fully encompass all aspects of the stress response observed in humans. Sex, age, and species are some of the basic experimental variables, but two other core aspects to consider are the duration and nature of the stressors.

In acute stress models, the stressor is only applied once and usually for a short time, while chronic models require repeated exposure to stressors over an extended period[37]. Regarding their nature, stressors can be coarsely divided into physical (eg, foot shock, forced swimming) and psychological (eg, social defeat, immobilization) stressors. Physical stressors usually entail potential bodily harm, while psychological stressors do not necessarily entail physical pain per se but rather the anticipation of physical pain, discomfort, or fear[37]. Depending on the nature of the stressor, different (and sometimes opposite) behavioral and physiological outcomes have been observed. For example, social defeat seems to increase high blood pressure while foot shock seems to decrease it[88]. Several authors have reviewed the question of how different types of stressors may favor the emergence of certain behavioral and physiological patterns[37,72]. In this study, we employed a chronic unpredictable stress protocol for 21 days. Longer protocols of chronic unpredictable stress have been used to induce depression-like behaviors and/or generate rodent models of depression since they seem to mimic the stress-induced depression observed in depressed patients[89-92]. In our study, we aimed to uncover the effect of chronic stress on striatal function and the emergence of stress-triggered motor changes. For that reason, we chose a protocol long enough to induce striatal alterations[12], but not necessarily aiming to bias the animal towards depressive-like phenotypes.

We exposed male mice once a day to one of three stressors: forced swimming, restraint, or social defeat. We only used male mice because social defeat is known to be difficult to implement in females since female mice are not naturally aggressive or strongly territorial[37,93]. Hence, for the time being, our observations and conclusions can be attributable to male mice only. Future work should focus on understanding sex-dependent susceptibility to chronic stress as well as its potential differential impact on specific striatal subregions such as DLS *versus* DMS and striosome *versus* matrix compartments of the striatum.

In summary, our study demonstrates that CS affects striatal GABAergic interneuron populations differently with SOM interneurons being particularly vulnerable. Although this study sheds light on the mechanistic link between CS and behavioral motor symptoms, it should be noted that the behavioral manifestation of prolonged stress exposure is a collection of separable behavioral symptoms that will likely rely on distinct brain areas and distinct neuronal microcircuits. Understanding how striatal circuits contribute to stress-induced motor symptoms will not solve the neural basis of anxiety, OCD, or PTSD in its entirety, but is certainly an important step towards establishing symptom-specific treatments. Hence, these findings may yield valuable insights for translational research in stress-related neuropsychiatric disorders.

## Methods

### Animals

All animal procedures were approved by local authorities Direção Geral de Alimentação e Veterinária (ID: DGAV 8519) and the Ethics Subcommittee for the Life Sciences and Health (SECVS) of University of Minho (ID: SECVS 01/18) and performed in accordance with European Community Council Directives (2010/63/EU) and the Portuguese law DL N° 113/2013 for the care and use of laboratory animals. Mice were group housed in a temperature-controlled room (22 °C; 55% humidity)

under a 12 h light/dark cycle (lights ON at 8 AM) with *ad libitum* access to water and food (4RF21, Mucedola). C57BL/6 mice were used for evaluating the physiological impact of chronic unpredictable stress, in vivo electrophysiology recordings, and proteomics analysis. *SOM-tdTomato* mice were generated by crossing *Sst-IRES-Cre* mice (JAX #013044)[94], which express Cre recombinase in somatostatin-expressing neurons, with *ROSA26-stopflox-tdTomato* cKI mice (gift from Dr. Guoping Feng and Dr. Fan Wang, MIT)[95]. *Pvalb-tdTomato* (JAX #027395)[96] and *SOM-tdTomato* mice were used for whole-cell patch clamp recordings. For opto-evoked postsynaptic currents recordings, *SOM-Cre* or *PV-Cre* (JAX #8069)[97] mice were crossed with *Ai27D* mice to ensure the expression of Channelrhodopsin-2 in SOM or PV interneurons, respectively. *SOM-Cre* mice were used for DREADD chemogenetics behavioral experiments. All mice lines (Table 1) were bread on a pure C57BL/6 background. Male mice were randomly assigned to the stress group with corresponding littermates assigned to the control (non-stressed) group and housed separately by the experimental group. Primers used for genotyping: *SOM-cre* (5′-TCT GAA AGA CTT G CG TTT GG-3′, 5′-TGG TTT GTC CAA ACT CAT CAA-3′ and 5′-GGG CC A GGA GTT AAG GAA GA-3′), *PV-cre* (5′-GCT CAG AGC CTC CAT TCC CT-3′, 5′-AGT ACC AAG CAG GCA GGA GAT ATC G-3′ and 5′-CAG CCT CTG TTC CAC ATA CAC TTC-3′), *Pvalb-tdTomato* (5′-ACT GCA GCG CTG GTC ATA TGA GC-3′ and 5′-ACT CTT TGA TGA CCT CCT CG-3′), *ROSA-tdTomato* (5′-CAC TTG CTC TCC CAA AGT CG-3′, 5′-TAG TCT AAC TCG CGA CAC TG-3′ and 5′-GTT ATG TAA CGC GGA ACT CC-3′) and Ai27D (5′-CCG AAA ATC TGT GGG AAG TC-3′, 5′-AAG GGA GCT GCA GTG GAG TA-3′, 5′-CTG TTC CTG TAC GGC ATG G-3′ and 5′-GGC ATT AAA GCA GCG TAT CC-3′).

### Chronic stress

Chronic unpredictable stress protocol was performed as previously described[12] with minor modifications. Briefly, 5 weeks old male mice were exposed once a day to one of three stressors: forced swimming, restraint or social defeat. The social defeat was based on the resident-intruder paradigm[98]. The intruder mouse was placed inside the resident mouse's cage and allowed to interact with the resident for a maximum of 5 min or until being attacked and defeated by the resident. Afterwards, the intruder was separated from the resident but kept inside the resident's cage for 30 min inside an acrylic enclosure that allowed visual, auditory, and olfactory contact but prevented further direct physical attack. Individually housed 3–12 months old male CD1 mice were used as residents. During forced swimming, mice were placed inside a 20 cm-diameter cylinder half-filled with 24 ± 1 °C water and forced to swim for 5 min. During restraint, mice were immobilized for 15 min inside a 50 mL falcon tube containing breathing holes. Stressors were randomly distributed throughout 21 days and arbitrarily scheduled in terms of daytime, to prevent animals from predicting and adapting to the stressor. This scheduled was conceived to maximize unpredictability and to better mimic the variability of stressors encountered in daily life. In all cohorts, mice were exposed to the same order and schedule of stressors. Body weights were monitored once a week and *post-mortem* thymus and adrenal weights were recorded.

### Immunohistochemistry

Mice were deeply anaesthetized with avertin (tribromoethanol; 20 mg/mL; dose of 0.5 mg/g body weight) by intraperitoneal injection and subsequently checked for lack of paw withdrawal reflexes before being transcardially perfused with saline followed by 4% paraformaldehyde (PFA). Brains were dissected out, post-fixed by overnight immersion in 4% PFA and then transferred to 30% sucrose in phosphate-buffered saline (PBS) solution for 24 h (immersion at 4 °C). Brains were then embedded in optimal cutting temperature compound (OCT; Bio-Optica) and 30 μm-thick coronal sections of the striatum were serially cut in the cryostat (Leica Microsystems). Sections were washed three

**Table 1 | Materials and methods**

| Reagent or resource | Source | Identifier |
|---|---|---|
| Virus and antibodies | | |
| AAV2-hSyn-DIO-hM4D(Gi)-mCherry | Addgene | 44362-AAV2 |
| AAV2-hSyn-DIO-hM3D(Gq)-mCherry | Addgene | 44361-AAV2 |
| Rabbit anti-FosB antibody | Cell signaling | 2251 |
| Rabbit anti-cFos antibody | Millipore | ABE457 |
| 488-Goat anti-rabbit IgG | Invitrogen | A11034 |
| Chemicals | | |
| CNO | Sigma | C0832 |
| PTX | Tocris | 1128 |
| TTX | Tocris | 1078 |
| NBQX | Tocris | 0373 |
| DL-AP5 | Tocris | 0105 |
| DAPI | Panreac Applichem | A1001,0010 |
| Mounting Medium Immu-Mount | ThermoFisher | 9990402 |
| Experimental Models: Mouse Strains | | |
| *Ssttm2.1(cre)Zjh/J* | Jackson Laboratory | 013044 |
| *B6;129P2-Pvalbtm1(cre)Arbr/J* | Jackson Laboratory | 8069 |
| *C57BL/6-Tg(Pvalb-TdTomato)15Gfng/J* | Jackson Laboratory | 027395 |
| *ROSA26-stopflox-tdTomato cKI* | Gift from Dr Fan Wang | 10.1371/journal.pone.0029423 |
| *B6.Cg-Gt(ROSA)26Sortm27.1(CAG-COP4\*H134R/tdTomato)Hze/J* | Jackson Laboratory | 012567 |
| *CD1 mice* | Charles River Laboratories | Crl:CD1(ICR) |
| *C57BL/6J mice* | Jackson Laboratory | 000664 |
| Software | | |
| Matlab | MathWorks | https://www.mathworks.com/products/matlab.html |
| Bonsai | Bonsai-RX | https://bonsai-rx.org/ |
| EthoVision XT 13 | Noldus | https://www.noldus.com/ethovision-xt |
| VAME | https://github.com/LINCellularNeuroscience/VAME | https://doi.org/10.1101/2020.05.14.095430 |
| DeepLabCut | https://github.com/DeepLabCut/DeepLabCut | https://www.nature.com/articles/s41593-018-0209-y |
| ProteinPilot™ | Sciex | https://sciex.com/ |
| PeakView™ | Sciex | https://sciex.com/ |
| pCLAMP | Molecular Devices | Clampfit 10.2 |
| Minianalysis | Synaptosoft | http://www.synaptosoft.com/MiniAnalysis/ |
| Prism | Graphpad | https://www.graphpad.com/ |
| Other | | |
| Nanoject II | Drummond Scientific | 3-000-204 |
| Point gray camera | FLIR Systems, Inc. | FL3-U3-13Y3M-C Flea3 |
| LED driver | Thorlabs | LEDD1B |
| Proteomics dataset PXD031193 | PRIDE repository | https://www.proteomexchange.org/ |

times for 10 min with PBS and placed in citrate buffer at 80 °C for 20 min. After that, brain sections were allowed to cool down at room temperature (RT) for 20 min and then washed 3 times for 10 min with PBS. Brain sections were permeabilized twice with 0.3% Triton X-100 (Sigma−Aldrich) in PBS for 10 min at RT. After washing 3 times in PBS for 10 min, brain sections were blocked using 15%NGS, 5%BSA, 0.2% Triton-X (blocking buffer) for 1 h at RT. Blocked sections were incubated overnight with primary antibody for FosB (Rabbit mAb #2251, Cell Signaling, 1:200) or cFos (ABE457, Millipore, 1:1000) diluted in blocking buffer. Following primary antibody incubation, brain sections were washed three times for 10 min in PBS and incubated with secondary antibody (488-Goat anti-rabbit IgG, A11034, Invitrogen, 1:1000), for 2 h at RT. Next, brain sections were washed three times for 10 min with PBS, stained for DAPI (Sigma−Aldrich) for 3 min at RT and mounted on Superfrost slides (Thermo Scientific) using Shandon™

Immu-Mount™ mounting medium (Thermo Scientific #9990402). Image acquisition was carried out using Olympus confocal microscope (FV1000, Olympus) and performed blinded to the experimental groups. FosB images were acquired with 20× objective and analyzed with ImageJ software[99] to count the number of DAPI nuclei, the number of cells expressing FosB and its intensity *per* cell. Four striatal slices (AP distance from bregma approximately between +1.2 and +0.6 mm) were used from each mouse. In each section three ROI areas (120 × 120 μm) were drawn within the dorsomedial striatum along the ventricle wall, on both hemispheres, and the number of cells expressing FosB was normalized for the number of DAPI nuclei inside the ROI.

**Synaptosomal preparation**
Proteomic analysis was performed from striatal synaptosomal samples. Briefly, mice were decapitated after avertin anesthesia

(tribromoethanol 20 mg/mL; dose of 0.5 mg/g body weight) and the striatum was microdissected and snap-frozen on liquid nitrogen. Striatal tissue from three mice (approximately 200 mg) was pooled together to generate one sample. All the buffers used for synaptosomal purification were supplied with protease inhibitor (cOmplete EDTA free, Roche) and phosphatase inhibitor (PhosSTOP, Roche). Striatal tissue was homogenized in 3 ml ice-cold buffer (4 mM HEPES pH 7.4, 0.32 M sucrose), at 4 °C using a mechanical tissue grinder by applying 30–40 strokes at 900 rpm. Homogenates were centrifuged for 15 min at 900 × $g$ at 4 °C, and supernatants were centrifuged again for 15 min at 900 × $g$ at 4 °C. The resulting supernatants were then centrifuged at 18,000 × $g$ for 15 min and the pellet was subsequently resuspended with 1.5 mL of HEPES buffer and centrifuged at 18,000 × $g$ for 15 min at 4 °C. The pellet was dissolved in 3 mL of hypo-osmotic buffer (4 mM HEPES, pH 7.4) and 8 manual strokes were applied. Then, the hypo-osmotic synaptosomal fraction was rotated for 1 h at 4 °C. Hypo-osmotic synaptosomal fractions were centrifuged for 20 min at 26,500 × $g$ at 4 °C and the pellets were dissolved in 200 μL buffer (50 mM HEPES pH 7.4, 2 mM EDTA) via sonication. Protein quantification was carried out using the BCA protein assay Kit from Biorbyt, according to the manufacturer's instructions.

## Proteomics

Short GeLC-SWATH-MS was used according to Anjo et al., 2015[100], with minor modifications. Briefly, 40 μg of each sample and a pooled sample per group were subjected to in-gel digestion after a partial SDS-PAGE run. LC-MS information was acquired in two different acquisition modes: information-dependent acquisition (IDA) of the pooled samples and SWATH (Sequential Windowed data independent Acquisition of the Total High-resolution Mass Spectra) acquisition of each individual sample. Protein identification and library construction were performed using ProteinPilot™ (v5.0.1, Sciex), and the relative quantification was performed using SWATH™ processing plug-in for PeakView™ (v2.2, Sciex). The mass spectrometry proteomics data have been deposited to the ProteomeXchange Consortium via the PRIDE[101] partner repository with the dataset identifier PXD031193. Further details can be found in the Sup. Information file.

## Electrophysiology

**Ex vivo recordings**. Acute striatal slices from control and chronic stressed mice were prepared as previously described[74,102]. Briefly, animals were deeply anesthetized with avertin (tribromoethanol; 20 mg/mL; Sigma–Aldrich) with a dose of 0.5 mg/g body weight by intraperitoneal injection and subsequently checked for lack of paw withdrawal reflexes. Mice were then transcardially perfused with 15–20 mL of carbogenated N-methyl-D-glucamine -based artificial cerebrospinal fluid (NMDG-aCSF) solution (mM): 92 NMDG, 2.5 KCl, 1.2 NaH$_2$PO$_4$, 30 NaHCO$_3$, 20 HEPES, 25 glucose, 5 sodium ascorbate, 2 thiourea, 3 sodium pyruvate, 10 MgSO$_4$.7H$_2$O and 0.5 CaCl$_2$.2H$_2$O, (7.2–7.4 pH and 300–310 mOsm/L). After decapitation, brains were rapidly removed and placed in the same carbogenated NMDG-aCSF solution for slice preparation. A Vibratome VT1000S (Leica Microsystems) was used to prepare 300 μm-thick striatum coronal slices. Slices were incubated at 32–34 °C for 11 min in carbogenated NMDG-aCSF solution and then transferred to a holding chamber (Brain Slice Keeper BSK4, Scientific Systems Design Inc.) filled with RT carbogenated aCSF solution (mM): 119 NaCl, 2.5 KCl, 1.2 NaH$_2$PO$_4$, 24 NaHCO$_3$, 12.5 glucose, 2 MgSO$_4$.7H$_2$O and 2 CaCl$_2$.2H$_2$O (7.2–7.4 pH and 300–310 mOsm/L). Slices were allowed to recover at least 1 h at RT before recordings. Recordings were made at RT (22–25 °C) and carbogenated aCSF was perfused at approximately 3 mL/min. Striatal cell-types were identified based on native fluorescence and recordings were obtained from cells with series-resistance values <25 MΩ. Patch pipettes were pulled from borosilicate-glass with filament (GB150F-8P, Science Products) on a P1000 horizontal puller (Sutter Instruments) with a typical resistance of 2–5 MΩ. For opto-evoked postsynaptic currents (oPSC), patch pipettes were backfilled with CsCl internal (mM): 103 CsCl, 12 CsOH, 12 methanesulfonic acid, 5 TEA-Cl, 10 HEPES, 4 MgATP, 0.3 Na$_2$GTP, 10 phosphocreatine, 0.5 EGTA, 5 lidocaine N-ethylchloride and 4 NaCl (pH adjusted to 7.3 with KOH and osmolarity adjusted to 300 mOsm/L with K$_2$SO$_4$). For recordings of miniature inhibitory post synaptic currents (mIPSCs), patch pipettes were backfilled with CsCl internal, cells were voltage-clamp at −70 mV, and slices were perfused with carbogenated aCSF in the presence of 50 μM DL-AP5 (dl-2-amino-5-phosphonovaleric acid, Tocris), 10 μM NBQX (2,3-Dioxo-6-nitro-1,2,3,4-tetrahydrobenzo[$f$]quinoxaline-7-sulfonamide, Tocris) and 1 μM tetrodotoxin (Tocris). mIPSC recordings were low-pass filtered at 2 KHz, digitized at 10 KHz, and analyzed using Minianalysis software (Synaptosoft). All recordings were performed in dorsomedial striatum (DMS) after seal rupture and internal equilibrium under a BX-51WI microscope (Olympus), equipped with fluorescence and infrared differential interference contrast (IR-DIC). Data was acquired using Digidata 1440 A and MultiClamp 700B (Molecular Devices, USA).

**In vivo recordings**. Extracellular activity was recorded from DMS (Bregma: AP +1.0, ML +/−1.5, DV −2.5 mm) under ketamine/medetomidine anesthesia (75 mg/Kg and 1 mg/Kg, respectively) with a 2-shank or 4-shank 64-channel probe (H2 or E1 respectively, Cambridge Neurotech) at 30 Ks/s with an Open Ephys acquisition system[103]. Spike sorting was performed with JRClust[104] with manual cluster curation based on inspection of spike waveforms, interspike interval histograms and auto- and cross-correlograms. Shape-based waveform parameters (half-width duration and through-to-peak duration) were calculated from each single-unit cluster mean waveform as in Gage et al., 2010[105] using custom-written Matlab code (Mathworks). Single-units were classified as putative MSNs or other cells (interneurons) according to clusters defined by a Gaussian Mixture Model fitted to the waveform parameters projection. Probes were coated with DiI (1,10-dioctadecyl-3,3,30,30-tetramethylindocarbocyanine perchlorate) (D3911, Thermo Fisher) prior to brain insertion to allow *post-mortem* confirmation of probe placement.

## Stereotactic injections

For chemogenetic manipulation of striatal SOM interneurons using DREADDs (Designer Receptors Exclusively Activated by Designer Drugs), male *SOM-cre* mice were stereotactically injected with AAV2-hSyn-DIO-hM3D(Gq)-mCherry (Addgene, 44361-AAV2, titer: 1.5 × 10$^{13}$ U/mL). Briefly, mice were deeply anesthetized with ketamine/medetomidine (75 mg/Kg and 1 mg/Kg, respectively), and placed in a small animal stereotaxic instrument (Kopf Instruments, Model 1900). 300 nL of concentrated virus solution was injected bilaterally in the DMS (Bregma: AP +1.0, ML +/−1.5, DV −2.5 mm) using a Nanoject device (Drummond Scientific; US) at low speed (100 nL/min). The injection tip was left in place for 5 min after the injection to allow proper diffusion and prevent backflow. Mice were allowed to recover for 4 days before starting the CS protocol. The injection site was confirmed in all animals by checking mCherry signal *post-mortem*.

## Behavior

All behavioral tests were performed at the end of CS protocol and during the light phase in a temperature and humidity-controlled room. For all procedures, mice were allowed to acclimate to the behavioral room for 30 min prior to starting the behavioral tests.

**Open field test (OF)**. Mice were tested in the OF for spontaneous exploration of a novel environment as described by[106]. Briefly, mice were individually placed in the brightly illuminated center of an open field arena (100–200 lux, 43 × 43 cm, Med Associates Inc., USA) and allowed to explore freely for 10 min. Data were collected using the

activity monitor software (Med Associates Inc., USA) and used to determine the total distance traveled (cm) and the percentage of total time spent in the center *versus* periphery areas of the OF.

**Chemogenetics.** CNO (Sigma Aldrich) was first dissolved in DMSO to a final concentration stock of 100 mM, and then diluted in physiological saline solution (0.9% NaCl) for individual injections. Male *SOM-Cre* mice expressing hM3D(Gq) in striatal SOM interneurons were intra-peritoneally injected either with saline or CNO (2 mg/kg), for two consecutive days, in which they received saline on day 1 and CNO on day 2. One hour after the injection, mice were placed in a custom transparent open field arena (35 × 35 cm) and filmed from the bottom with a video camera (FL3-U3-13Y3M-C Flea3 Camera, FLIR Systems, Inc.) for 10 min. Mice were tracked in real-time using Bonsai[107,108], and videos were acquired at 60 fps for subsequent markerless pose estimation with DeepLabCut. c-Fos immunohistochemistry was checked *post mortem* to confirm neuronal activation triggered by CNO injection.

**Pose estimation and behavioral motifs classification.** Body part tracking for pose estimation was performed using DeepLabCut (DLC, version 2.2.0.1)[62,109]. Briefly, 100 frames taken from 48 videos of mice freely exploring the OF were manually labeled (95% of the frames were used for model training). Next, a ResNet-50-based convolutional neural network was used with default parameters for $1 \times 10^6$ number of training iterations (obtained test error: 4.32 pixels; train error: 3.29 pixels; image size: 890 × 990 pixels). Trained network was then used to analyze videos from similar experimental settings. Unsupervised classification of behavioral motifs was performed with VAME 1.0[63] using body parts position estimates from videos analyzed with the previously trained DLC model. VAME's recurrent neural network model was trained with 40 videos ($1.65 \times 10^6$ frames) and the resulting multivariate signal was clustered into 38 behavioral motifs with k-means clustering. Various k values ranging from 20 to 40 were tested for k-means clustering. As assessed by visual inspection of representative videos for each cluster, $k = 38$ created the most homogeneous clusters, i.e., clusters representing clear individual behaviors and not a mix of different behaviors. To avoid overclustering and to group similar behavioral motifs, a hierarchical representation of the clusters was created based on the transition probability between motifs. By pruning the resulting hierarchical tree, 8 behavioral communities were identified post-hoc: walk, turn, rear, stationary rotation, stationary exploration/pause/sniff, groom, dig, and other. The latter cluster included different behaviors that could not be easily identified with a specific behavioral label. Behavioral communities were confirmed by visual inspection of the Uniform Manifold Approximation and Projection (UMAP) embedding representation of the communities and representative videos of manually labeled post-hoc communities. Motif/Community usage was calculated as the number of frames each mouse spent on each behavioral motif/community.

### Statistical analysis
Statistical analyses were performed using GraphPad Prism 9 software (GraphPad Software Inc.) and Matlab (Mathworks). Statistical details can be found in figure legends.

### Reporting summary
Further information on research design is available in the Nature Portfolio Reporting Summary linked to this article.

## Data availability
The mass spectrometry proteomics data generated in this study have been deposited to the ProteomeXchange Consortium via the PRIDE[101] partner repository with the dataset identifier PXD031193 (http://www.proteomexchange.org/). All the data generated in this study are provided in the Supplementary Information/Source Data file. Source data are provided with this paper.

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

## Acknowledgements

We thank all lab members for their continuous support and helpful discussions. Research in the Monteiro laboratory was funded by The Branco Weiss fellowship Society in Science, the European Molecular Biology Organization (EMBO) Long-Term Fellowship (ALTF 89–2016), and Fundação para a Ciência e a Tecnologia (FCT) grant number PTDC/MED-NEU/28073/2017 and POCI-01-0145-FEDER-028073. D.R. was supported by FCT doctoral fellowship reference PD/BD/137759/2018 through the Inter-University Doctoral Program in Ageing and Chronic Disease (PhDOC). L.J. is supported by "la Caixa" Banking Foundation under the grant agreement LCF/PR/HR21-00410. A.C. is supported by FCT doctoral fellowship 2022.11778.BD. C.S. was supported by Ph.D. fellowship SFRH/BD/88419/2012, co-financed by the European Social Fund (ESF) through POCH - Programa Operacional do Capital Humano and national funds via FCT. B.M. would like to acknowledge the support from the European Regional Development Fund (ERDF), through the COMPETE 2020 - Operational Program for Competitiveness and Internationalization and Portuguese national funds via FCT, under projects: POCI-01-0145-FEDER-30943 (ref.: PTDC/MEC-PSQ/30943/2017), POCI-01-0145-FEDER-016428 (ref.: SAICTPAC/0010/2015). The National Mass Spectrometry Network (RNEM) provided funding under the contract POCI-01-0145-FEDER-402-022125 (ref.: ROTEIRO/0028/2013), and UIDB/04539/2020 and UIDP/04539/2020. F.M. would like to acknowledge the support from NORTE-01-0145-FEDER-000039, supported by Norte Portugal Regional Operational Program (NORTE 2020), under the PORTUGAL 2020 Partnership Agreement, through the European Regional Development Fund (ERDF) and National funds, through FCT project UIDB/50026/2020 and UIDP/50026/2020. P.M. is supported by FEBS (Federation of European Biochemical Societies) Excellence Awards 2021, FCT 2021.01032.CEECIND and FCT 2022.05228.PTDC. We thank scidraw.io for illustrations.

## Author contributions

D.R. performed the experiments with help from L.J., M.F., A.C.C., A.C., and P.M. C.S. and B.M. performed proteomics experiment with help from D.R. and P.M. D.R., L.J., and P.M. designed and performed the analyses. D.R. wrote the manuscript with help by L.J., and P.M. N.S., F.M., and P.M. contributed materials and scientific feedback, and commented on the manuscript.

## Competing interests

The authors declare no competing interests.
