## [Peer Review File · Nature Communications]

Chronic stress causes striatal disinhibition mediated by SOM-interneurons in male miceREVIEWER COMMENTS

Reviewer #1 (Remarks to the Author):

This is an interesting study on the effect of chronic stress on striatal function related to GABAergic control of MSNs and associated behaviors. The individual experiments are well-conducted but there is a significant number of concerns regarding the flow of logic and aspects of brain diseases that are modeled.

1) A case is made for modeling aspects of OCD, but the behavioral components of the study are limited to the OF. This is not enough. In general, there is a confusion between what is being modeled: stress and multiple brain disorders are mentioned, without making a strong case for investigating the relevant pathway and without making a full case for the best assays and behaviors.

2) The “remarkable heterogeneity” that is noticed in behavior and physiology may well be due to the assays rather than the mice. The ROC approach is interesting, but still relying only on a short set of tests that are known to be variable.

3) The proteomic results are quite non-specific and do not necessarily point to presynaptic GABAergic neuron changes.

4) In vivo ephys- few interneurons detected in the stressed mice. The numbers of interneurons recorded are very small. How do we know that the group difference is not a random finding?

5) DREADD studies. This was done in stressed mice. The absence of non stressed mice precludes the replication of the first findings (i.e reduced activity in center), so we do not know whether activating SOM cells "reverses" the phenotype in this cohort. i.e., we can't compare behaviours between CS induced and SOM activation

6) The analysis of behavioral motif is quite interesting but it is surprising that it is not included upfront in the results section, as a stronger bases for the follow-up analyses. As such it feels a little bit like a “fishing exploration” for association with behavioral components.

7) There is no clear mention of the sex of the animals. Only one instance towards the end of the animal section suggest that the CUS study was only performed in male mice. Were females included? If not, it is required to justify why female mice were excluded, and it is critical to mention in the discussion that this observation, and conclusion can only be attributable to males, for the time being.

8) Not clear if all ~180 mice from the Control and CUS groups were tested at the same time, or through multiple cohorts.

9) Also unclear what subset of animals was used for proteomics (decapitated) or for IHC (perfused), and

how they were selected. This is of particular importance, in particular for the Synaptosomal preparation where 3 mouse samples were pooled to generate 1 final sample.

10) Figure 2 and Figure 3: It is important to clarify whether what is referred as Control and CS animals in Figure 2 and 3, is the “regular” groups as they were initially Control or CS, or if they follow the new group affiliation past ROC. This is important since some animals may have changed group. Maybe another terminology for those would be better Control (for “True Control”) and tCS maybe.

11) “chronic unpredictable stress that minimizes resilience” provides reference for this statement

12) What is the correlation structure of the outcome measures (OF, weight, thymus) and what is known of their direct correlation with altered stress response?

MINOR

- OF: lacks details allowing reproducibility, such as light intensity in the room, of particular importance since the author conclude of potential increased anxiety-phenotype.
- Chronic stress used is only for 21 days – authors should discuss the choice of such duration, when other protocol uses longer exposure time (5-6-7 weeks).
- Consistency: authors use CS or CUS for chronic stress. Please homogenize according to general chronic stress (CS) or to their own use of chronic stress paradigm (CUS).
- Line 39: Reduced body weight gain should be considered as a % of baseline.
- Unclear why there are different N between Fig1A-B-C and D. Please explain.
- Line 172-174: Way of administration of CNS should be specified there, to prevent readers from flipping pages to find the information in the method section.
- Fig 5: absence of Control CNO is a limitation that should be discussed.

Reviewer #2 (Remarks to the Author):

Rodrigues et al perform a multilevel examination of the effects of chronic stress on striatal activity. The manuscript is very informative. The manuscript is well written and the figures are really clear and easy to understand. The authors use two models of chronic stress: swim test and social defeat, which add to the rigor of reproducibility of the manuscript. Authors combine state of art methods: behavior, electrophysiology, histology, proteomics, and chemo-genetics to examine the effect of stress. Collectively authors reach a very clear conclusion that SOM interneurons of DMS are selectively affected by chronic stress.

I think that paper may benefit from additional two discussion paragraphs:

1) SOM interneurons do not act in isolation. SOM and PV inter-neurons are part of cortical-striatal circuit. That is possible that the cortico-SOM connection is affected by stress differently than the cortical-PV connection. For example, our own work demonstrates that stress in comparison to Huntington's disorder has different effects on cortico-PV-striatal circuits (Friedman et al 2017 and 2020).

Stress results in cortical "disconnection" from PV, HD results in PV disconnection from striosomal MSNs. Important to think in this context about the different computational roles of PV and SOM interneurons. Also, Levin's group demonstrates that in the case of HD PV and SOM interneurons are affected in different ways (see refs). Reading this paper my hypothesis is stress causes disconnection of cortex from PV and affects SOM interneurons activity as described in the manuscript.

2) In addition may be important to discuss different chronic stress paradigms, like chronic immobilization or foot-shock vs social defeat or swim test.

A) For example in Levin's paper, this group has multiple publications about differential effect of som and PV interneurons. Striatal GABAergic interneuron dysfunction in the Q175 mouse model of Huntington's disease.

B) example of computational work

Wilson: Division and subtraction by distinct cortical inhibitory networks in vivo.

C) Example of PV computational role (also has good set of PV /SOM computational papers).

Striosomes Mediate Value-Based Learning Vulnerable in Age and Huntington's Model

Reviewer #3 (Remarks to the Author):

This work aims at deciphering the mechanisms underlying stress-induced motor dysfunction. Indeed, chronic stress exposure is associated with a number of neuropsychiatric disorders such as obsessive-compulsive disorder (OCD), posttraumatic stress disorder (PTSD), anxiety and depression but it also worsens the progression of motor-associated conditions notably Parkinson's and Huntington diseases. The authors use an elegant set of neuroproteomics and functional (electrophysiology, optogenetics, chemogenetics) experiments combined with probabilistic machine learning to analyze in detail animal behaviors to demonstrate that chronic stress affects the function of GABAergic striatal neurons. This brain region was targeted considering its important role in stress responses and motor function. Results reported suggest that chronic stress exposure leads to overactivation of striatal neurons by reducing the connectivity between GABAergic somatostatin (SOM)-positive interneurons and medium spiny neurons (MSN). Thus, targeting striatal SOM interneurons may represent an interesting therapeutic target to modulate stress-related motor symptoms observed in neurodegenerative diseases and psychiatric disorders.

Overall, this is a well written, novel, interesting study. Figures are particularly well designed and compelling. The individual physiological phenotype scoring is an intriguing way to compile results and highlight individual differences, this is appreciated. I do have a number of concerns and suggestions as listed below that need to be addressed before I can recommend publication.

Major:

- Sex of the animals should be mentioned in the abstract and throughout the main text. Unpredictable stress is known to induce a depressive phenotype in female mice sooner vs males.

- Is locomotion affected in a non-anxious environment? Addition of locomotor tests in a non-anxious environment would strengthen the conclusion that manipulation of striatal SOM interneurons is beneficial under chronic stress conditions only.
- If I do understand well the rationale unstressed control animals with D-score of 2 (about 20%) as established in Fig.1 were included in the following brain analyzes of the stressed group? Can we still talk about stress effect then?
- FosB is not exclusively expressed by neurons but also glial cells. Was it considered in the Fig.3 analysis? This should be mentioned as a limitation.
- Could lack of effect of light stimulation of ChR2-expressing parvalbumin neurons in stressed mice be due to a ceiling effect with an increased of mIPSC frequency already observed in Fig.4B?
- Are optogenetically-evoked postsynaptic currents in SOM neurons far from physiological values (Fig.4N)?
- Unstressed controls for saline and CNO treatments should be added to the chemogenetic experiments in Fig.5. Comparison of stressed CNO vs stressed saline animals do not allow for direct assessment if the maladaptive behavior is reversed or not by manipulating SOM interneurons activity with normalization being performed on a stressed group.
- The splash test could also be added to confirm changes in self-grooming observed with DeepLabCut.

Minor:

- The type of stressors used in the chronic unpredictable stress paradigm should be listed in the main text as the protocols vary between labs and inclusion of social defeat is uncommon.
- Is the circulating level of corticosterone affected by chronic stress exposure?
- Only one time point after stress was studied so I do not understand the reference to temporal cumulative effects of chronic stress in the striatum (line 124). This appears to be extrapolated vs the data provided.

REVIEWER'S COMMENTS

We are truly grateful to all reviewers' positive comments and particularly to their insightful comments and suggestions that undoubtedly strengthen our manuscript. As you can see from the details below, these suggestions have led to performing many very interesting new experiments and to the rethinking of some of our data. We believe that the manuscript is significantly improved and we thank the reviewers for this process.

Reviewer #1 (Remarks to the Author):

This is an interesting study on the effect of chronic stress on striatal function related to GABAergic control of MSNs and associated behaviors. The individual experiments are well-conducted but there is a significant number of concerns regarding the flow of logic and aspects of brain diseases that are modeled.

1) A case is made for modeling aspects of OCD, but the behavioral components of the study are limited to the OF. This is not enough. In general, there is a confusion between what is being modeled: stress and multiple brain disorders are mentioned, without making a strong case for investigating the relevant pathway and without making a full case for the best assays and behaviors.

Author response: We thank the reviewer for this comment and apologize for the confusion and misunderstanding. Our study focuses on the effect of chronic stress on striatal function. Functionally, the striatum coordinates multiple aspects of motor and action planning. Notably, dysfunction of striatal circuits have been reported under chronic stress (DOI: 10.1038/nn.4087; DOI: 10.1126/science.1171203; DOI: 10.1016/j.cell.2017.10.017) and chronic stress exposure is a well-recognized triggering factor for a number of neuropsychiatric disorders, namely obsessive-compulsive disorder (OCD; characterized by repetitive motor routines), posttraumatic stress disorder (PTSD; characterized by hyperarousal and exaggerated startle reactions), but it also worsens the progression of motor-associated conditions, namely Parkinson's and Huntington diseases. Uncovering the pathway by which stress triggers motor dysfunction might be relevant to explain why motor symptoms emerge / worsen in multiple brain disorders after chronic stress exposure. Following reviewers' suggestion, we have now made major changes to improve the logic flow of the manuscript, starting with the abstract. We also added extra behavioral experiments relevant to stress-induced motor dysfunction that, we believe, significantly improved and strengthened our conclusions.

2) The "remarkable heterogeneity" that is noticed in behavior and physiology may well be due to the assays rather than the mice. The ROC approach is interesting, but still relying only on a short set of tests that are known to be variable.

Author response: We thank the reviewer for this comment and could not agree more: behavioral tests are known to be variable, making it difficult to conclude whether heterogeneity would emerge from the assay rather than the mice. This is exactly why we modified the ROC approach published by Cerniauskas et al, 2019 (<https://doi.org/10.1016/j.neuron.2019.09.005>). In that paper, authors use the results from 3 different behavioral tests to construct ROC curves to classify stressed animals: Elevated plus maze (EPM), Sucrose preference test (SPT), and Tail suspension test (TST). Here, we modified such approach by constructing ROC curves that are not based on behavioral assays but rather based on physiological parameters known to be affected by stress: body weight gain, adrenals, and thymus weights. Because these parameters

are less prone to assay variability, the ROC classification method becomes more objective and reproducible across labs.

We have now included the following sentence in the discussion to make it clear for the reader:

“One of the major challenges hindering the study of stress is the remarkable variability in individual responses and vulnerability to chronic stress. To address this issue, we developed a physiological-based ROC classification approach, based on the work from Cerniauskas et al, 2019. Because behavioral assays (previously used by Cerniauskas et al. to construct ROC curves) can be highly variable across laboratories (depending on experimental behavioral settings, housing conditions, etc), using physiological parameters as we have done here (body weight gain, adrenals, and thymus weights) to construct ROC curves, may be a more suitable approach to reduce variability, helping to improve the classification method and reproducibility.”

3) *The proteomic results are quite non-specific and do not necessarily point to presynaptic GABAergic neuron changes.*

Author response: We agree with the reviewer that proteomics can be relatively non-specific *per se*, requiring other finer approaches to reach solid conclusions. Although later in the manuscript we use a set of functional (electrophysiology, optogenetics, chemogenetics) experiments combined with animal behavior to dissect how chronic stress affects striatal circuits, it was the proteomics data that led us down the road of investigating the population of GABAergic interneurons (rather than neurons) in the striatum. Among the top 10 downregulated proteins revealed by proteomics after CS exposure, five of those proteins were GABAergic interneurons' proteins. Although such results do not necessarily point to presynaptic GABAergic neuron changes, this was the seminal finding that led us to focus on that class of cells when investigating in detail the consequences of CS over striatal circuits, being later confirmed by the functional experiments performed in this manuscript.

4) *In vivo ephys- few interneurons detected in the stressed mice. The numbers of interneurons recorded are very small. How do we know that the group difference is not a random finding?*

Author response: Indeed, this is a technical limitation of *in vivo* anesthetized recordings in striatum where spontaneous firing rate is low and the population of interneurons represents only ~5% of the total population of cells (please see DOI 10.1016/j.neuron.2008.11.005 and also 10.1146/annurev.neuro.051508.135422). Nonetheless, our percentage of recorded single units corresponding to interneurons is in accordance with typical numbers reported in the literature (especially in DMS region, please see *Berke et al*, Fig.1C,D, DOI: 10.1016/j.neuron.2004.08.035, and *Friedman et al*, Figure S4, DOI: 10.1016/j.cell.2017.10.017). Despite being a technical limitation of *in vivo* anesthetized recordings in striatum, we agree that the group difference previously observed could be a random finding induced by the very low number of interneurons detected in stressed mice (n=4 interneurons, out of 77 single units in stress group). Following the reviewer's suggestion, we have now performed new experiments and succeeded to isolate 292 single units in controls and 312 single units in stressed mice. Among these single units, 16 are interneurons in controls and 13 in stress, which is in accordance with the percentages reported in the literature (5.2% in controls and 4.0% in stress). These new interneurons' data reveals again reduced firing rate in stressed mice; a new Fig.3 is copied below.

5) DREADD studies. This was done in stressed mice. The absence of non stressed mice precludes the replication of the first findings (i.e reduced activity in center), so we do not know whether activating SOM cells "reverses" the phenotype in this cohort. i.e., we can't compare behaviours between CS induced and SOM activation.

Author response: We thank the reviewer for this comment. Despite the difficulties in performing such challenging experiments (obtaining enough heterozygous SOM-cre+ mice born at the same time, selecting only male mice, using matched littermate controls, successful viral expression of hM3 bilaterally in DMS after surgeries, 21-days of chronic stress protocol, ROC curve classification for obtaining true control and true stress mice that then qualify for subsequent behavioral experiments), we agree that such extra mile would be needed if we want to disentangle between CS induced vs SOM activation effects. Following reviewer's suggestion, we have repeated this experiment but this time using both a stressed and a non-stressed group, both injected with saline and CNO. Again, we found increased locomotor activity after 21 days of chronic stress, but we did not find differences in terms of center/periphery ratio. Despite surprising (reduced OF center/periphery ratio is a well-known finding in the stress field and also a phenotype replicated in our hands as it can be seen in Fig.1), such result might likely be explained by the fact that our chemogenetic behavioral testing is done in a custom arena adapted for filming the mice from the bottom but likely not suitable as an OF arena for interpreting the time spent in the center vs periphery of the arena. One of the key aspects behind the classical OF testing is the fact that rats and mice display a natural aversion to the brightly lit open area in the center of the OF. Here, however, the chemogenetic arena is evenly illuminated throughout and slightly smaller than our commercial OF testing arena from Med Associates. In other words, although the chemogenetic arena may be useful to study patterns of motor activity, caution should be applied when interpreting the distanced travelled in the center vs periphery of this arena. Indeed, in our previous version of the manuscript we did not find differences in the center/periphery ratio of stressed animals when treated with saline vs CNO in this arena. But this new data further reveals that there are no differences even between WT and stressed animals (nor between

any saline and CNO), meaning that no conclusions can be drawn in regard to this measure. Importantly, in line with the results obtained in Fig.1, as well as the results obtained in our previous version of the manuscript, we once again found that stressed mice have an hyperlocomotion phenotype when compared with controls (increased total distance) and that activation of SOM cells in stressed mice is sufficient to reduce locomotion to WT levels. Because of the stringent experimental control groups now employed, we are very confident on these results and are grateful to all reviewers' suggestions.

6) The analysis of behavioral motif is quite interesting but it is surprising that it is not included upfront in the results section, as a stronger bases for the follow-up analyses. As such it feels a little bit like a “fishing exploration” for association with behavioral components.

Author response: We thank the reviewer for this comment. When we started this study 5 years ago we did not have such an advanced behavioral analysis pipeline, hence, we grouped and analyzed our ~180 mice based on the only common measurements that we had available at the time. However, we do agree that for the new cohorts it would be useful to include analysis of behavioral motifs. As such, our new Fig.5 now includes 38 behavioral motifs for mice from all groups, including control, stress, saline and CNO conditions. The new data is now included in Fig.5:

7) *There is no clear mention of the sex of the animals. Only one instance towards the end of the animal section suggest that the CUS study was only performed in male mice. Were females included? If not, it is required to justify why female mice were excluded, and it is critical to mention in the discussion that this observation, and conclusion can only be attributable to males, for the time being.*

Author response: Although the importance of understanding sex-dependent susceptibility to chronic stress is undeniable for the development of more appropriate therapeutic approaches, in our study we only used male mice. We decided to perform our study only in male C57B6J mice because it is known that the vulnerability to chronic stress and the neuronal and behavioral responses to stress are sex dependent (Refs: 10.1016/j.yhbeh.2006.06.033; 10.1016/j.biopsycho.2004.11.009; 10.2307/2095420; 10.1111/ejn.15481; 10.7554/eLife.11352.002; 10.1523/JNEUROSCI.1392-15.2015; 10.1016/j.biopsycho.2009.02.007; 10.1093/cercor/bhq003; 10.3389/fnins.2013.00092;). Thus, to avoid potential biological heterogeneity of the results, we decided not to include females when we first designed this study. The other reason that made us select males only was the fact that most chronic stress studies in mice (including the 21-day CUS protocol that we use in this manuscript), have been validated in males only. One of the stressors included in our 21-day CUS protocol is social defeat which is known to be difficult to implement in females since female mice are not naturally aggressive or strongly territorial (eliciting male aggression against a female of reproductive age is very difficult; please see ref 10.1016/j.neubiorev.2017.01.037). Only recently have there been attempts to develop an effective protocol of social defeat for use in females: Refs 10.1016/j.biopsycho.2019.05.005; 10.1016/j.biopsycho.2019.08.007; 10.1038/npp.2017.259. Lastly, the phase of female estrous cycle may influence behavior and their response to chronic stress exposure (please see ref. 10.3389/fnmol.2019.00074; and ref. 10.1038/s41598-019-48683-3), which would require synchronization of the estrous cycle between all the females, further complicating our already complex study design. However, we fully agree with the reviewer that the lack of clear information about the sex of the animals throughout the manuscript may confuse the reader. We have now revised the manuscript to clearly state the sex of the animals. Furthermore, we now make it clear in the discussion that our findings are exclusive to male mice only.

8) *Not clear if all ~180 mice from the Control and CUS groups were tested at the same time, or through multiple cohorts.*

Author response: We thank the reviewer for this comments that further adds to the rigor and reproducibility of our manuscript. Our ~180 mice displayed in Fig.1 were tested through multiple cohorts and combined to provide sufficient power to perform the ROC curve analysis. Although not tested at the same time, all cohorts were composed by 5-weeks old male mice and littermates were randomly assigned to the control vs stress group in a balanced manner. Following the reviewer's question, we have decided to investigate all the multiple cohorts and check whether any particular cohort(s) could be skewing the group average data. The results have now been included as Sup.Fig.1 and show all physiological parameters (adrenals, thymus and body weights) grouped *per* cohort, rather than individual data points *per* mice as displayed in Fig.1. Each cohort has been color coded as a way to provide visual information and help clarifying the reader. Individual cohorts show similar trends as averaged data from the total mice and no skewing of the data was observed towards any particular cohort(s).

9) Also unclear what subset of animals was used for proteomics (decapitated) or for IHC (perfused), and how they were selected. This is of particular importance, in particular for the Synaptosomal preparation where 3 mouse samples were pooled to generate 1 final sample.

Author response: Mice were randomly assigned to proteomics or IHC cohorts and balanced littermates were included in the control and stress groups. Body weight, adrenals and thymus weights were collected from all animals to perform ROC curve analysis and classify each mouse as a true Control or true Stress animal before proteomics. In the stress group, only animals with a D-score of 2 or 3 were used for further experiments (animals from the stress group with a D-score of 0 or 1 were excluded from proteomics or any other experiment). In the control group, only animals with a D-score of 0 or 1 were used for further experiments (animals from the control group with a D-score of 2 or 3 were excluded from proteomics or any other experiment). As per reviewer's request we have included below the details pertaining to the specific subset of animals combined for the final proteomic samples (individual mouse samples pooled to generate 1 final sample are color coded):

Figure - Color-coded representation of individual mice that were combined in the same sample tube for proteomic analysis.

(Left) Body weight gain for each control and stress mouse used for proteomic analysis, after 21 days of CS protocol. n=12 control and n=12 chronic stress mice; All animals pooled together to generate a single striatum sample are represented in the same color (total = 4 control samples and 4 stress samples).

(Middle) Adrenal glands weight normalized to individual body weight for each control and stress mouse used for proteomic analysis, after 21 days of CS protocol. Each colored circle represents the same sample tube as in panel A. n=12 control and n=12 chronic stress mice.

(Right) Thymus weight normalized to individual body weight for each control and stress mouse used for proteomic analysis, after 21 days of CS protocol. Each colored circle represents the same sample tube as in panel A and B. n=12 control and n=12 chronic stress mice.

10) *Figure 2 and Figure 3: It is important to clarify whether what is referred as Control and CS animals in Figure 2 and 3, is the “regular” groups as they were initially Control or CS, or if they follow the new group affiliation past ROC. This is important since some animals may have changed group. Maybe another terminology for those would be better Control (for “True Control”) and tCS maybe.*

Author response: We thank reviewer 1 for raising this question and would like to start by clarifying that none of the animals have changed group or gained a new group affiliation after ROC classification.

In Fig.1, all mice subjected to the stress protocol were considered part of the stressed group. But after ROC curve classification (Fig.2 onwards), only stressed animals with a D-score of 2 or 3 were used in the manuscript (“true stress”). Any mice from the stress group with a D-score of 0 or 1 were excluded from further analysis without any re-assignment / new group affiliation.

The same rule applies to the control group: only animals with a D-score of 0 or 1 were used in this study (“true control”) and all other animals from the control group with a D-score of 2 or 3 were excluded from further experiments. To make this clearer, we have now added the following paragraph (bold) in the manuscript:

“(…) We therefore proceeded only with animals from the stress group if they had a D-score of 2 or 3 (“true” stressed mice), and from the control group if they had a D-score of 0 or 1 (“true” controls). No animals have been reassigned / changed group or have been included in any of the subsequent analyzes if not meeting the above criteria.”

Furthermore, we have now changed Figure 1 designation to “Control group” vs “Stress group”, whereas after ROC classification (Fig.2 onwards) the designation is “control” vs “stress”, making it clear for the reader that this is no longer the initial bulk group but rather individual mice truly classified as control or stress mice.

11) *“chronic unpredictable stress that minimizes resilience” provides reference for this statement*

The following references have been included: DOI: 10.1126/science.1171203, DOI: 10.1038/nn1399 and DOI 10.1016/j.crneur.2021.100013

The sentence has also been rephrased to better cite the corresponding literature:
“This protocol of chronic unpredictable stress aims to avoid the resilient effect of behavioral control over stressors”

12) *What is the correlation structure of the outcome measures (OF, weight, thymus) and what is known of their direct correlation with altered stress response?*

Author response: We thank the reviewer for this excellent question. Although several behavioral alterations have been reported in parallel with physiological changes after chronic stress exposure (for example, some studies have linked adrenal glands weight and plasma corticosterone, a stress hormone, to social dominance/status [10.3181/00379727-94-23067; 10.1016/s0016-6480(67)80006-6; 10.1016/s0003-3472(72)80179-9; 10.1016/0031-9384(84)90047-7; 10.1016/j.physbeh.2016.12.038], which is also deeply intertwined with stress response [10.1016/j.ynstr.2014.10.004]), a clear correlation structure between such physiological changes and altered stress

response often remains elusive. In order to directly address the reviewer's question, we have investigated the correlation structure of our outcome measures (OF, body weight, thymus, adrenals), both in the control and stress group. As shown in the correlation matrix below, none of the outcome measures are correlated with each other (CONTROL: thymus vs body $R^2=0.005$; adrenals vs body $R^2=0.015$; OF vs body $R^2=0.04$; adrenals vs thymus $R^2=0.02$; OF vs thymus $R^2=8.7\times 10^{-5}$; OF vs adrenals $R^2=0.001$; STRESS: thymus vs body $R^2=0.04$; adrenals vs body $R^2=0.02$; adrenals vs thymus $R^2=0.004$; OF vs body $R^2=2.1\times 10^{-5}$; OF vs thymus $R^2=0.0004$; OF vs adrenals $R^2=0.002$).

Given that none of these measures are individually correlated, we then asked the question whether combining all of them could allow us to correctly guess the animal group (control or stress). To answer this question, we applied a supervised machine learning algorithm (Support Vector Machine, SVM) to check whether we could achieve good classification accuracy when these physiological measures were used together to train the classifier. The resulting SVM classifier accuracy was 63% (average of 100 repetitions, holding out 20% of randomly selected samples in each repetition for testing). This means that when all physiological measures (thymus, adrenals and body weights) are combined together to predict the animal's group, the classifier can assign each animal into the stress or control group correctly in approx. 60% of the cases. Although performing above chance-level, the classifier's accuracy was low. Next, we performed ROC curve analysis and reran the SVM classifier in correct D-score animals only. Notably, the accuracy of the SVM classifier increased to 82% after D-scoring classification, meaning that our ROC curve approach helps improving classification methods and correctly assigns animals as true stress or true controls in 82% of the cases. The figure below shows the accuracy distribution for the 100 repetitions of the classifier training/testing before (blue) and after (orange) D-scoring, and has now been included in Sup.Fig.3:

MINOR

- *OF: lacks details allowing reproducibility, such as light intensity in the room, of particular importance since the author conclude of potential increased anxiety-phenotype.*

Author response: Information has been added on the methods section:

Open Field test

“(...) mice were individually placed in the brightly illuminated centre of an open field arena (100-200 lux, 43 × 43 cm, Med Associates Inc., USA)”

- *Chronic stress used is only for 21 days – authors should discuss the choice of such duration, when other protocol uses longer exposure time (5-6-7 weeks).*

Author response: As mentioned by the reviewer, we use a chronic unpredictable stress protocol for 21 days, whereas other groups may use longer exposure time. The longer protocols of chronic unpredictable stress have been used in the literature to induce depression-like behaviors and/or generate rodent models of depression since they seem to mimic the stress-induced depression observed in depressed patients (please see refs 10.1007/s002130050456; 10.1038/sj.mp.4001457; 10.1002/0471142301.ns0810as55; and 10.1007/978-1-61779-458-2_6). In our study, we aim to uncover the effect of chronic stress on striatal function and the emergence of stress-triggered motor changes. For that reason, we chose a protocol long enough to induce striatal alterations, but not necessarily aiming to bias the animal towards depressive-like phenotypes. We have now included this topic in the Discussion section.

- *Consistency: authors use CS or CUS for chronic stress. Please homogenize according to general chronic stress (CS) or to their own use of chronic stress paradigm (CUS).*

Author response: We apologize for the inconsistency. All abbreviations have now been homogenized to chronic stress (CS).

- *Line 39: Reduced body weight gain should be considered as a % of baseline.*

Author response: We would like to thank the suggestion that we have carefully considered. Our rationale to use the raw data is because we think those values might be useful for other groups that want to reproduce our data and use our ROC classification method. We have therefore decided to maintain raw values in this manuscript.

- *Unclear why there are different N between Fig1A-B-C and D. Please explain.*

Author response: Our ~180 mice displayed in Fig.1 were tested through multiple cohorts throughout this project and combined to provide sufficient power to perform the ROC curve analysis. Although these cohorts were not tested all at the same time, we have thymus, adrenals and body weight data from all those mice (n=88 control and 90 CS, Fig1A-B-C) and OF data for nearly all of them (n=83 control and 84 CS, Fig1D).

- *Line 172-174: Way of administration of CNS should be specified there, to prevent readers from flipping pages to find the information in the method section.*

Author response: As suggested by the reviewer, this information has been added to main text:

“After 21 days of stress exposure, control and stress mice were placed in an open arena to examine the impact of chemogenetic activation of hM3D(Gq)-expressing SOM interneurons by intraperitoneal injection of saline or CNO”

- *Fig 5: absence of Control CNO is a limitation that should be discussed.*

Author response: Following the question raised by the reviewer in comment 5) “DREADD studies”, we have repeated this experiment which now includes a Control CNO group. Please see reply above (comment 5 “DREADD studies”).

Reviewer #2 (Remarks to the Author):

Rodrigues et al perform a multilevel examination of the effects of chronic stress on striatal activity. The manuscript is very informative. The manuscript is well written and the figures are really clear and easy to understand. The authors use two models of chronic stress: swim test and social defeat, which add to the rigor of reproducibility of the manuscript. Authors combine state of art methods: behavior, electrophysiology, histology, proteomics, and chemo-genetics to examine the effect of stress. Collectively authors reach a very clear conclusion that SOM interneurons of DMS are selectively affected by chronic stress.

I think that paper may benefit from additional two discussion paragraphs:

- 1) SOM interneurons do not act in isolation. SOM and PV inter-neurons are part of cortical-striatal circuit. That is possible that the cortico-SOM connection is affected by stress differently than the cortical-PV connection. For example, our own work demonstrates that stress in comparison to Huntington's disorder has different effects on cortico-PV-striatal circuits (Friedman et al 2017 and 2020). Stress results in cortical "disconnection" from PV, HD results in PV disconnection from striosomal MSNs. Important to think in this context about the different computational roles of PV and SOM interneurons. Also, Levin's group demonstrates that in the case of HD PV and SOM interneurons are affected in different ways (see refs). Reading this paper my hypothesis is stress causes disconnection of cortex from PV and affects SOM interneurons activity as described in the manuscript.*

- 2) In addition may be important to discuss different chronic stress paradigms, like chronic immobilization or foot-shock vs social defeat or swim test.*

A) For example in Levin's paper, this group has multiple publications about differential effect of som and PV interneurons. Striatal GABAergic interneuron dysfunction in the Q175 mouse model of Huntington's disease.

B) example of computational work

Wilson: Division and subtraction by distinct cortical inhibitory networks in vivo.

C) Example of PV computational role (also has good set of PV /SOM computational papers).

Striosomes Mediate Value-Based Learning Vulnerable in Age and Huntington's Model

Author response: We are very humbled by this positive assessment of our manuscript and particularly grateful to the reviewer's extremely insightful comments. Inspired by these lines of thought, our discussion has now been completely rewritten and reflects more deeply on the implications of our work in light of the existing literature. This process has significantly improved our manuscript and we are very thankful to the reviewer for such a constructive reviewing process. The new discussion is included below.

DISCUSSION

Our findings demonstrate that chronic exposure to stress leads to overactivation of striatal circuits by reducing the connectivity between GABAergic somatostatin (SOM)-positive interneurons and medium spiny neurons (MSN). One of the major challenges hindering the study of stress is the remarkable variability in individual responses and vulnerability to chronic stress. To address this issue, we developed a physiological-based ROC classification approach based on the work from *Cerniauskas et al*, 2019⁴¹, to more confidently identify which animals were indeed stressed. Because behavioral assays (previously used by *Cerniauskas et al.* to construct ROC curves) can be highly variable across laboratories (depending on experimental behavioral settings, housing conditions, etc), using physiological parameters to construct ROC curves, as we have done here, may be a more suitable approach to reduce variability, helping to improve the classification method and reproducibility. Leveraging on this classification approach, we then studied the effect of chronic stress (CS) on striatal function and showed that DMS (dorsomedial striatum) is particularly vulnerable to CS, corroborating previous findings from imaging studies performed in stressed rats and humans^{18,65}.

Vulnerability of striatal GABAergic interneurons to chronic stress

Our proteomic results raise the possibility of GABAergic interneurons being major targets of chronic stress in the striatum. The striatum contains two major interrelated classes of GABAergic interneurons: the parvalbumin (PV) and somatostatin (SOM) positive interneurons. Together these two cell types provide strong inhibitory control of striatal output^{56,57} and a tight orchestration between them gives rise to the spatial and temporal properties of local activity⁶⁶. The importance of such cooperation is well illustrated in a recent study whereby using a single interneuron type (PV) in a spiking network model does not allow modulation of excitatory and inhibitory firing rates independently, as it occurs *in vivo*⁶⁶. The authors demonstrate that strong feedback from SOM interneurons is required for this independent modulation to occur. Thus, disrupting the activity of either SOM or PV has profound consequences on global network firing activity.

In the striatum, SOM and PV interneurons are part of the cortical-striatal circuit: cortex excites both PV and SOM GABAergic interneurons, causing inhibition of local MSNs. Recent work from *Friedman et al.* showed that the cortical-PV excitatory connection is weakened in chronically stressed animals. Furthermore, the authors hypothesize that the PV-MSN inhibitory connection is likely spared¹⁴. Our results corroborate this hypothesis: by optogenetically activating PV-interneurons and directly recording from connected MSNs, no changes are detected in optogenetically-evoked postsynaptic currents (oPSC) between stress and control animals, suggesting that the PV-MSN connection is indeed intact after chronic stress. By contrast, we show that SOM-MSN connection is severely affected in stressed animals with significantly reduced oPSCs. This suggests that stress differently impacts both interneuron types in the striatum through different mechanisms (cortex-PV *versus* SOM-MSN) but with the same outcome: striatum disinhibition. In other words, stress leads to cortical “disconnection” from PVs (indirectly weakening their inhibitory control over MSNs) and also “disconnects” SOM from MSNs, directly weakening their inhibitory control over MSNs. The two mechanisms potentiate striatum disinhibition under stress.

A causal link between dysfunction of SOM cells and stress-related neuropsychiatric disorders has been previously hypothesized based on rodent studies showing that: 1) chronic stress induces low RNA levels of somatostatin in corticolimbic regions⁶⁷; 2)

brain-wide silencing of SOM cells elevates anxiety-like behaviors ⁶⁸; and 3) brain wide disinhibition of SOM cells through genetic inactivation of GABAA receptors has antidepressant-like effects ⁶⁸. What makes striatal SOM (but not PV) interneurons direct targets of chronic stress is still an open question. In the hippocampus, for example, it has been shown that SOM interneurons express high levels of glucocorticoid receptors (GRs), being well positioned to respond directly to circulating glucocorticoids, while only distinct populations of PV interneurons are direct targets ⁶⁹. However, careful consideration should be taken when extrapolating findings from one brain region to another. Although GR immunoreactivity is present in nearly 90% of striatal cells, SOM interneurons in the striatum do not seem to contain glucocorticoid receptor immunoreactivity ⁷⁰. This suggests that stress likely exerts its effects over SOM interneurons through different mechanisms in hippocampus *versus* striatum regions.

Considering such lessons from different brain regions, it is perhaps important to keep in mind that the dorsal striatum itself is composed by two similar yet functionally distinct regions: the dorsolateral “sensorimotor” striatum (DLS) and the dorsomedial “associative” striatum (DMS). Chronic stress causes opposing structural changes in these two regions and promotes a shift from flexible “cognitive” states (DMS encoded) to more rigid “habit” states (DLS encoded) ^{12,71}, rapidly switching the brain from reflective to reflexive control of behavior ⁷². Interestingly, a region-specific bias exists in the action of striatal GABAergic interneurons: while PV interneurons control MSN activity more efficiently in the DLS, SOM interneurons more efficiently control MSNs activity in DMS ⁷³. Furthermore, the relative density of PV expression is known to decrease from DLS to DMS ^{74,75}, which may explain why SOM-MSN connectivity, but not PV-MSN connectivity, is primarily affected in DMS after chronic stress.

The effect of chronic stress on striatal function and its relevance to disease

Despite scientific evidence linking stress to striatal dysfunction and to motor symptoms worsening / triggering in neuropsychiatric disorders ^{22,27,28,76}, the effect of chronic stress on striatal function remains elusive. Deficits in interneurons’ GABAergic transmission, with consequent overactivation/ disinhibition of striatum circuits has been proposed as one key mechanism for motor symptoms in OCD and Tourette syndrome ⁷⁷⁻⁸², two disorders known to be affected by chronic stress ^{19,76,83-85}.

Our *in vivo* electrophysiology recordings reveal reduced activity of local interneurons and abnormally increased MSN activity in DMS (homologous to the human caudate) after chronic stress, mimicking findings of increased caudate activity in OCD patients ^{32,33} but also corroborating findings from other rodent models of CS ^{14,86,87}.

One of the most replicated findings in human OCD studies is increased activity in the orbitofrontal and anterior cingulate/caudal medial prefrontal cortex, which are also the main source of input to the striosomal system in the caudate nucleus. Notably, MSNs located at the striosome compartment seem to be particularly vulnerable to chronic stress ¹⁴, further suggesting a link between stress, dorsomedial striatum (caudate) and neuropsychiatric disorders ¹¹⁻¹⁸. A common mechanistic explanation would be that chronic stress releases striatum from the inhibitory influence of local interneurons, promoting unusual motor behavioral output. In our current work, using chemogenetics and probabilistic machine learning to analyze the latent structure of spontaneous animal behavior, the results demonstrate that *in vivo* modulation of SOM interneurons in DMS region selectively alters motor activity of chronically stressed male mice.

Experimental limitations and future perspectives

Despite serving as valuable tools for studying stress physiology, animal models of stress can never fully encompass all aspects of the stress response observed in humans. Sex, age, and species are some of the basic experimental variables, but two other core aspects to consider are the duration and nature of the stressors.

In acute stress models, the stressor is only applied once and usually for a short time, while chronic models require repeated exposure to stressors over an extended period³⁷. Regarding their nature, stressors can be coarsely divided into physical (eg, foot shock, forced swimming) and psychological (eg, social defeat, immobilization) stressors. Physical stressors usually entail potential bodily harm, while psychological stressors do not necessarily entail physical pain *per se* but rather the anticipation of physical pain, discomfort, or fear³⁷. Depending on the nature of the stressor, different (and sometimes opposite) behavioral and physiological outcomes have been observed. For example, social defeat seems to increase high blood pressure while foot shock seems to decrease it⁸⁸. Several authors have reviewed the question of how different types of stressors may favor the emergence of certain behavioral and physiological patterns^{37,72}. In this study, we employed a chronic unpredictable stress protocol for 21 days. Longer protocols of chronic unpredictable stress have been used to induce depression-like behaviors and/or generate rodent models of depression since they seem to mimic the stress-induced depression observed in depressed patients^{89–92}. In our study, we aimed to uncover the effect of chronic stress on striatal function and the emergence of stress-triggered motor changes. For that reason, we chose a protocol long enough to induce striatal alterations¹², but not necessarily aiming to bias the animal towards depressive-like phenotypes.

We exposed male mice once a day to one of three stressors: forced swimming, restraint, or social defeat. We only used male mice because social defeat is known to be difficult to implement in females since female mice are not naturally aggressive or strongly territorial^{37,93}. Hence, for the time being, our observations and conclusions can be attributable to male mice only. Future work should focus on understanding sex-dependent susceptibility to chronic stress as well as its potential differential impact on specific striatal subregions such as DLS *versus* DMS and striosome *versus* matrix compartments of the striatum.

In summary, our study demonstrates that CS affects striatal GABAergic interneuron populations differently with SOM interneurons being particularly vulnerable. Although this study sheds light on the mechanistic link between CS and behavioral motor symptoms, it should be noted that the behavioral manifestation of prolonged stress exposure is a collection of separable behavioral symptoms that will likely rely on distinct brain areas and distinct neuronal microcircuits. Understanding how striatal circuits contribute to stress-induced motor symptoms will not solve the neural basis of anxiety, OCD, or PTSD in its entirety, but is certainly an important step towards establishing symptom-specific treatments. Hence, these findings may yield valuable insights for translational research in stress-related neuropsychiatric disorders.

Reviewer #3 (Remarks to the Author):

This work aims at deciphering the mechanisms underlying stress-induced motor dysfunction. Indeed, chronic stress exposure is associated with a number of neuropsychiatric disorders such as obsessive-compulsive disorder (OCD), posttraumatic stress disorder (PTSD), anxiety and depression but it also worsens the progression of motor-associated conditions notably Parkinson's and Huntington diseases. The authors use an elegant set of neuroproteomics and functional (electrophysiology, optogenetics, chemogenetics) experiments combined with probabilistic machine learning to analyze in detail animal behaviors to demonstrate that chronic stress affects the function of GABAergic striatal neurons. This brain region was targeted considering its important role in stress responses and motor function. Results reported suggest that chronic stress exposure leads to overactivation of striatal neurons by reducing the connectivity between GABAergic somatostatin (SOM)-positive interneurons and medium spiny neurons (MSN). Thus, targeting striatal SOM interneurons may represent an interesting therapeutic target to modulate stress-related motor symptoms observed in neurodegenerative diseases and psychiatric disorders.

Overall, this is a well written, novel, interesting study. Figures are particularly well designed and compelling. The individual physiological phenotype scoring is an intriguing way to compile results and highlight individual differences, this is appreciated. I do have a number of concerns and suggestions as listed below that need to be addressed before I can recommend publication.

Author response: We very much thank the reviewer for this overall positive assessment of our manuscript and we are also grateful for the constructive criticism to improve our manuscript. We have carefully considered all concerns and suggestions and performed many more key experiments to address all the points. We believe our manuscript is significantly strengthened and hope the reviewer can now recommend publication.

Major:

- Sex of the animals should be mentioned in the abstract and throughout the main text. Unpredictable stress is known to induce a depressive phenotype in female mice sooner vs males.

Author response: We thank the reviewer's suggestion and have now revised the manuscript to clearly state the sex of the animals. Additionally, we now make it clear in the discussion that, for the time being, our observations and conclusions can be attributable to male mice only.

- Is locomotion affected in a non-anxious environment? Addition of locomotor tests in a non-anxious environment would strengthen the conclusion that manipulation of striatal SOM interneurons is beneficial under chronic stress conditions only.

Author response: Excellent question. Following reviewer's suggestion, and to specifically address the reviewer's question on whether the manipulation of striatal SOM interneurons is beneficial under chronic stress conditions only, we have decided to repeat Fig.5 experiment but now in a non-anxious environment (homecage). Our new results reveal that stressed mice do not seem to display hyperlocomotion in a non-anxious environment (homecage total distance travelled: control vs stress $p=0.72$). However, activation of SOM interneurons in the homecage environment suggests a trend

to decrease locomotion but without significant differences (Stress saline vs Stress CNO $p=0.17$). The new results are shown below and have been included as Sup.Fig.6.

- If I do understand well the rationale unstressed control animals with D-score of 2 (about 20%) as established in Fig.1 were included in the following brain analyzes of the stressed group? Can we still talk about stress effect then?

Author response: We thank the reviewer for raising this question and would like to start by clarifying that none of the animals have been included in the following brain analyzes if not meeting the proper D-score criteria for its own original group.

As such, in Fig.1 (before D-scoring), all mice subjected to the stress protocol were considered part of the stressed group. But after ROC curve classification (Fig.2 onwards), only stressed animals with a D-score of 2 or 3 were used in the manuscript ("true stress"). Any mice from the stress group with a D-score of 0 or 1 were excluded from further analysis without any re-assignment / new group affiliation.

The same rule applies to the control group: only animals with a D-score of 0 or 1 were used in this study ("true control") and all other animals from the control group with a D-score of 2 or 3 were excluded from further experiments. To make this clearer, we have now added the following paragraph (bold) in the manuscript:

"(...) We therefore proceeded only with animals from the stress group if they had a D-score of 2 or 3 ("true" stressed mice), and from the control group if they had a D-score of 0 or 1 ("true" controls). No animals have been reassigned/changed group or have been included in any of the subsequent analyzes if not meeting the above criteria."

- FosB is not exclusively expressed by neurons but also glial cells. Was it considered in the Fig.3 analysis? This should be mentioned as a limitation.

Author response: We thank the reviewer for raising this important aspect. We did not consider glial cells in our initial analysis but after reviewer's suggestion, we decided to perform additional IHC experiments and perform co-staining for FosB and NeuN (a neuronal nuclear antigen) in DMS slices from control and stressed animals. Our results show approx. ~90% colocalization between FosB and NeuN signal in both controls and stressed mice and the data has now been included as Sup.Fig.4. Although these results indicate that the majority of FosB is being expressed by neurons in the DMS and that the results from the original manuscript are related mainly to FosB expression in neurons, we cannot discard that glial cells might also be participating in the process although perhaps in smaller scale.

- Could lack of effect of light stimulation of ChR2-expressing parvalbumin neurons in stressed mice be due to a ceiling effect with an increased of mIPSC frequency already observed in Fig.4B?

Author response: We thank the reviewer for raising this interesting possibility. Indeed, if the mIPSCs in Fig.4B were being recorded from MSNs, this could be one possibility. We note, however, that our reduced mIPSC frequency was recorded from PV interneurons (Fig.4B) and our ChR2-PV light stimulation recordings were obtained from downstream target MSNs (please see intracellular pipette positioning schematics in Fig.4 panel A versus K). Nonetheless, we have wondered about the possibility raised by the reviewer, and performed mIPSC frequency recordings from MSNs to investigate that possibility, but no differences were found.

- Are optogenetically-evoked postsynaptic currents in SOM neurons far from physiological values (Fig.4N)?

Author response: The values in such experiments are likely not representative of the true physiological values because CsCl internal solution is used to increase the driving force, making it possible to record mIPSC at -70mV holding potential. Nonetheless, recordings were obtained in the same conditions for all experimental groups and the observed values match those reported in the literature for SOM-optogenetically-evoked postsynaptic currents in MSNs (approx. 500 pA, see paper from Sabatini lab, 2016, Fig.1D, <https://www.ncbi.nlm.nih.gov/pmc/articles/PMC5074692/>). However, we do agree that the representative trace initially used by us in Fig.4M was probably not the best example as it represented one of the largest events but not one of the most usually recorded. We have now replaced it with an example trace that is much more representative of most oPSC amplitude (~500pA).

- Unstressed controls for saline and CNO treatments should be added to the chemogenetic experiments in Fig.5. Comparison of stressed CNO vs stressed saline animals do not allow for direct assessment if the maladaptive behavior is reversed or not by manipulating SOM interneurons activity with normalization being performed on a stressed group.

Author response: We thank the reviewer for this comment. Despite the difficulties in performing such challenging experiments (obtaining enough heterozygous SOM-cre+ mice born at the same time, selecting only male mice, using matched littermate controls,

successful viral expression of hM3 bilaterally in DMS after surgeries, 21-days of chronic stress protocol, ROC curve classification for obtaining true control and true stress mice that qualify for subsequent behavioral experiments), we agree that such extra mile would be needed for truly assessing if the maladaptive behavior can be reversed or not. Following reviewer's suggestion, we have now repeated this experiment but this time including unstressed controls for saline and CNO treatments. The new results are now included as Fig.5 and we would like to thank the reviewer for this suggestion that really improved our manuscript and helped strengthening our conclusions.

- The splash test could also be added to confirm changes in self-grooming observed with DeepLabCut.

Following reviewer's suggestion, we have now performed splash test followed by manual behavior scoring and confirmed changes in self-grooming as observed previously with DeepLabCut. Furthermore, despite being a small pilot cohort (n=4 control and n=4 stress), we were able to observe a trend for increased grooming in the stress group after CNO treatment, a trend that was not observed in the control group. The new results are shown below and have been included in the manuscript as Sup.Fig.5.

Minor:

- The type of stressors used in the chronic unpredictable stress paradigm should be listed in the main text as the protocols vary between labs and inclusion of social defeat is uncommon.

Author response: Information has been added in the main text as per reviewer's request.

- Is the circulating level of corticosterone affected by chronic stress exposure?

Author response: We thank the reviewer for this question. Indeed, circulating levels of corticosterone are often used to evaluate stress levels in rats. In mice, however, many studies do not report this persistent increase of circulating corticosterone levels, perhaps due to the low blood sample volumes that can be drawn without disturbing the animals, making it difficult to accurately quantify corticosterone levels by the standard assays (such as ELISA). However, given what is known from the rat's literature, we also initially collected blood samples for our mice and quantified its circulating corticosterone levels as suggested by the reviewer. Although being a good predictor in stressed rats, our results revealed that CORT was not a good predictor for mice in our hands, with its ROC curve basically matching the chance level curve. Such results are likely due to technical limitations in accurately detecting circulating CORT levels in mice, rather than CORT not being a relevant stress marker in mice. Notably, a trend for increase can still be observed in the average stress group data. Results are now included in Sup.Fig.2:

- Only one time point after stress was studied so I do not understand the reference to temporal cumulative effects of chronic stress in the striatum (line 124). This appears to be extrapolated vs the data provided.

Author response: We apologize for the misunderstanding. Although only one time point after stress was studied, we are looking to the cumulative effect of 21-days of chronic stress (hence the referral to “temporal cumulative effects”) as opposed to studies focused on acute stress. We have now rephrased the manuscript to avoid confusing the readers.

REVIEWERS' COMMENTS

Reviewer #2 (Remarks to the Author):

The authors address all the critiques. And manuscript will be a great addition to our understanding role of striatal interneurons under stress

Reviewer #3 (Remarks to the Author):

The authors addressed all my concerns and suggestions. I can now recommend publication and congratulate them for this elegant body of work.